# DEEP UNLEARNING: FAST AND EFFICIENT TRAINING-FREE APPROACH TO CONTROLLED FORGETTING

## ABSTRACT

Machine *unlearning* has emerged as a prominent and challenging area of interest, driven in large part by the rising regulatory demands for industries to delete user data upon request and the heightened awareness of privacy. Existing approaches either retrain models from scratch or use several finetuning steps for every deletion request, often constrained by computational resource limitations and restricted access to the original training data. In this work, we introduce a novel class unlearning algorithm designed to strategically eliminate an entire class or a group of classes from the learned model. To that end, our algorithm first estimates the Retain Space and the Forget Space, representing the feature or activation spaces for samples from classes to be retained and unlearned, respectively. To obtain these spaces, we propose a novel singular value decomposition-based technique that requires layer wise collection of network activations from a few forward passes through the network. We then compute the shared information between these spaces and remove it from the forget space to isolate class-discriminatory feature space for unlearning. Finally, we project the model weights in the orthogonal direction of the class-discriminatory space to obtain the unlearned model. We demonstrate our algorithm's efficacy on ImageNet using a Vision Transformer with only $\sim 1.5\%$ drop in retain accuracy compared to the original model while maintaining under $1\%$ accuracy on the unlearned class samples. Further, our algorithm consistently performs well when subject to Membership Inference Attacks showing $7.8\%$ improvement on average across a variety of image classification datasets and network architectures, as compared to other baselines while being $\sim 6\times$ more computationally efficient. Additionally, we investigate the impact of unlearning on network decision boundaries and conduct saliency-based analysis to illustrate that the post-unlearning model struggles to identify class-discriminatory features from the forgotten classes.

## 1 INTRODUCTION

Machine learning has automated numerous applications in various domains, including image processing, language processing, and many others, often surpassing human performance. Nevertheless, the inherent strength of these algorithms, which lies in their extensive reliance on training data, paradoxically presents potential limitations. The literature has shed light on how these models behave as highly efficient data compressors (Tishby & Zaslavsky, 2015; Schelter, 2020), often exhibiting tendencies toward the memorization of full or partial training samples (Arpit, 2017; Bai et al., 2021). Such characteristics of these algorithms raise significant concerns about the privacy and safety of the general population. This is particularly concerning given that the vast training data, typically collected through various means like web scraping, crowdsourcing, user data collection through apps and services, and more, is not immune to personal and sensitive information. The growing awareness of these privacy concerns and the increasing need for safe deployment of these models have ignited discussions within the community and, ultimately, led to some regulations on data privacy, such as Voigt & Von dem Bussche (2017); Goldman (2020). These regulations allow the use of the data with the mandate to delete personal information pertaining to a user if they choose to opt-out from sharing their data. The mere deletion of data from archives is not sufficient due to the memorization behavior of these models. This necessitates machine unlearning algorithms that can remove the influence of requested data or unlearn those samples from the model. A naive approach, involving the retraining of models from scratch, guarantees the absence of information from sensitive samples but is often impractical, especially when dealing with compute intensive State-of-The-Art

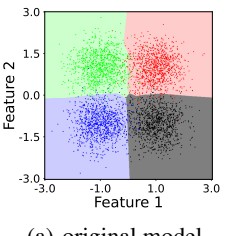 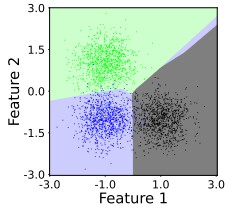 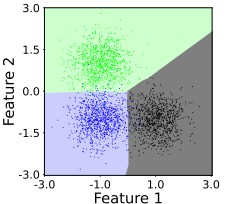

| (a) original model | (b) model forgetting Red Class | (c) Retrained model |

**Figure 1:** Illustration of the unlearning algorithm on a simple 4 class classification problem. Figure shows the decision boundary for (a) original model, (b) our unlearnt model redistributing the space to nearby classes and (c) Retrained model without red class.

(SoTA) models like ViT(Dosovitskiy et al., 2020). Further, efficient unlearning poses considerable challenges, as the model parameters do not exhibit a straightforward connection to the training data (Shwartz-Ziv & Tishby, 2017). Moreover, these unlearning algorithms may only have access to a fraction of the original training data, further complicating the unlearning process.

Our work focuses on challenging scenarios of class unlearning and multi-class unlearning (task unlearning) (Golatkar et al., 2020a; 2021). For a class unlearning setup, the primary goal of the unlearning algorithm is to eliminate information associated with a target class from a pretrained model. This target class is referred to as the forget class, while the other classes are called the retain classes. The unlearning algorithm should produce parameters that are functionally indistinguishable to those of a model trained without the target class. The key challenges in such unlearning is three folds (i) pinpointing class-specific information within the model parameters, (ii) updating the weights in a way that effectively removes target class information without compromising the model's usability on other classes and (iii) demonstrating scalability on large scale dataset with well trained models. The current SoTA for class unlearning (Tarun et al., 2023) shows acceptable accuracy on the retained classes compared to the original model, achieving minimal unlearning times. However, the authors present their results on undertrained models having $\sim 10\%$ lower accuracy for the original model on the entire dataset. The memorization behavior of the model is generally exhibited during the later training stages where the model overfits the training data (Feldman & Zhang, 2020) and it might not be fair to evaluate unlearning when the model is not trained to convergence. Additionally, the results of this work are presented only on small datasets like CIFAR10 and CIFAR100 (Krizhevsky et al., 2009). The practical use of such algorithms may be limited by their performance when applied to well-trained models on large-scale datasets. In this work, we ask the question *"Can we unlearn class (or multiple classes) from a well trained model given access to few samples from the training data on a large dataset?"*. Having a few samples is particularly interesting if the unlearning algorithms have to efficiently scale to large datasets having many classes to ensure fast and resource efficient unlearning algorithm.

We draw insights from work by Saha et al. (2021) in the domain of continual learning, where the authors use the Singular Value Decomposition (SVD) technique to estimate the gradient space essential for the previous task and restrict future updates in this space to maintain good performance on previous learning tasks. This work demonstrates a few samples ( about 125 samples per task ) are sufficient to obtain a good representation of the gradient space. Our work proposes to strategically eliminate the class discriminatory information by updating the model weights to maximally suppress the activations of the samples belonging to unlearn class. We first estimate the Retain Space and the Forget Space, representing the feature or activation spaces for samples from classes to be retained and unlearned, respectively. We propose a novel singular value decomposition-based technique to obtain these spaces, which requires layer wise collection of network activations from a few samples through the network. We then compute the shared information between these spaces and remove it from the forget space to isolate class-discriminatory feature space for unlearning. Finally, we project the model weights in the orthogonal direction of the class-discriminatory space to obtain the unlearned model. We demonstrate our algorithm on a simple 4 way classification problem with input containing 2 features as shown in Figure 1. The decision boundary learnt by the trained model is shown in Figure 1(a) while the model unlearning the red class exhibits the decision boundary depicted in Figure 1(b). The decision boundary for a retrained model is shown if Figure 1(c) This illustration shows that the proposed algorithm redistributes the input space of the class to be unlearnt

to the closest classes. The experimental details of this demonstration are provided in Appendix A.1. Our algorithm demonstrates SOTA performance in class unlearning setup with access to very few samples from the training dataset ( less than $4\%$ for all our experiments). As our algorithm relies on very few samples from the train dataset it efficiently scales to large datasets like ImageNet (Deng et al., 2009), where we demonstrate the results using 1500 samples ($0.00117\%$ of the training dataset).

In summary, the contributions of this work are listed as follows,

- We propose a novel Singular Value decomposition based class unlearning algorithm which uses very few samples from the training data and does not rely on gradient based optimization steps. To the best of our knowledge, our work is the first to demonstrate class unlearning results on ImageNet for SoTA transformer based models.
- We evaluate our algorithm on various datasets and with a variety of models and show our algorithm consistently outperforms the State of the Art methods. Additionally, we provide evidence that our model's behavior aligns with that of a model trained without the forget class samples through membership inference attacks, saliency-based feature analyses and confusion matrix analyses.
- We demonstrate the applicability of our algorithm to two practical scenarios in multi-class unlearning: (i) One-shot Multi-Class unlearning ( or task unlearning) through a single step of multi-class unlearning and (ii) Sequential Multi-Class unlearning through multiple steps of single class unlearning, demonstrating the capability of processing multiple unlearning requests over time.

## 2   RELATED WORKS

**Unlearning:** Many unlearning algorithms have been introduced in the literature, addressing various unlearning scenarios, including item unlearning (Bourtoule et al., 2021), feature unlearning (Warnecke et al., 2021), class unlearning (Tarun et al., 2023), and task unlearning (Parisi et al., 2019). Some of these solutions make simplifying assumptions on the learning algorithm. For instance, Ginart et al. (2019) demonstrate unlearning within the context of k-mean clustering, Brophy & Lowd (2021) present their algorithm for random forests, Mahadevan & Mathioudakis (2021) and Izzo et al. (2021) propose an algorithm in the context of linear/logistic regression. Further, there have been efforts in literature, to scale these algorithms for convolution layers Golatkar et al. (2020a;b). Note, however, the algorithms have been only demonstrated on small scale problems. In contrast, other works, such as Bourtoule et al. (2021), suggest altering the training process to enable efficient unlearning. This approach requires saving multiple snapshots of the model from different stages of training and involves retraining the model for a subset of the training data, effectively trading off compute and memory for good accuracy on retain samples. Unlike these works our proposed algorithm does not make any assumptions on the training process or the algorithm used for training the original model.

**Class Unlearning:** The current State-of-the-Art (SoTA) for class unlearning is claimed by Tarun et al. (2023). In their work, the authors propose a three stage unlearning process, where the first stage learns a noise pattern that maximizes the loss for each of the classes to be unlearnt. The Second stage (also called the impair stage) unlearns the class by mapping the noise to the forget class. Finally, if the impair stage is seen to reduce the accuracy on the retained classes, the authors propose to finetune the impaired model on the subset of training data in the repair stage (the third stage). This work presents the results on small datasets with undertrained models and utilizes up to $20\%$ of the training data for the unlearning process. Further, the work by Chundawat et al. (2023) proposes two algorithms which assumes no access to the training samples. Additionally, authors of Baumhauer et al. (2022) propose a linear filtration operator to shift the classification of samples from unlearn class to other classes. These works lose considerable accuracy on the retain class samples and have been demonstrated on small scale datasets like MNIST and CIFAR10. Our work demonstrates results on SoTA Vision transformer models for the ImageNet dataset, showing the effective scaling of our algorithm on large dataset with the model trained to convergence.

**Other Related Algorithms :** SVD is a well known technique used to constrain the learning in the direction of previously learnt tasks in the continual learning setup Saha et al. (2021); Chen et al. (2022); Saha & Roy (2023). These methods are sample efficient in estimating the gradient space relevant to a task. Recent work by Li et al. (2023) proposes subspace based federated unlearning using SVD. The authors perform gradient ascent in the orthogonal space of input gradient spaces formed by other clients to eliminate the target client's contribution in a federated learning setup.

Such ascent based unlearning is generally sensitive to hyperparameters and is susceptible to catastrophic forgetting on retain samples. As our proposed approach does not rely on such gradient based training steps it is less sensitive to the hyperparameters. Moreover, the techniques could be used on top of our method to further enhance the unlearning performance.

## 3 PRELIMINARIES

**Class Unlearning:** Let the training dataset be denoted by $\mathcal{D}_{train} = \{(x_i, y_i)\}_{i=1}^{N_{train}}$ consisting of $N_{train}$ training samples where $x_i$ represents the network input and $y_i$ is the corresponding label. The test dataset be $\mathcal{D}_{test} = \{(x_i, y_i)\}_{i=1}^{N_{test}}$ containing $N_{test}$ samples. Consider a function $y = f(x_i, \theta)$ with the parameters $\theta$ that approximates the mapping of the inputs $x_i$ to their respective labels $y_i$. In the case of a well-trained deep learning model with parameters $\theta$, there would be numerous samples $(x_i, y_i) \in \mathcal{D}_{test}$ for which the relationship $y_i = f(x_i, \theta)$ holds. For a class unlearning task aimed at removing the target class $t$, the training dataset can be split into two partitions, namely the retain dataset $\mathcal{D}_{train\_r} = \{(x_i, y_i)\}_{i=1; y_i \neq t}^{N}$ and the forget dataset $\mathcal{D}_{train\_f} = \{(x_i, y_i)\}_{i=1; y_i=t}^{N}$. Similarly the test dataset can be split into these partitions as $\mathcal{D}_{test\_r} = \{(x_i, y_i)\}_{i=1; y_i \neq t}^{N}$ and $\mathcal{D}_{test\_f} = \{(x_i, y_i)\}_{i=1; y_i=t}^{N}$. The objective of the class unlearning algorithm is to derive unlearnt parameters $\theta_f^*$ based on $\theta$, a subset of the retain partition $\mathcal{D}_{train\_sub\_r} \subset \mathcal{D}_{train\_r}$, and a subset of the forget partition $\mathcal{D}_{train\_sub\_f} \subset \mathcal{D}_{train\_f}$. The parameters $\theta_f^*$ must be functionally indistinguishable from a network with parameters $\theta^*$, which is retrained from scratch on the samples of $\mathcal{D}_{train\_r}$ in the output space. In other words, these parameters must satisfy $f(x_i, \theta^*) \simeq f(x_i, \theta_f^*)$ for $(x_i, y_i) \in \mathcal{D}_{test}$ or $\mathcal{D}_{train}$.

**SVD:** A rectangular matrix $A \in \mathbb{R}^{d \times n}$ can be decomposed using SVD as $A = U\Sigma V^T$ where $U \in \mathbb{R}^{d \times d}$ and $V \in \mathbb{R}^{n \times n}$ are orthogonal matrices and $\Sigma \in \mathbb{R}^{d \times n}$ is a diagonal matrix containing singular values Deisenroth et al. (2020). The columns of matrix $U$ are the $d$ dimensional orthogonal vectors sorted by the amount of variance they explain for $n$ samples (or the columns) in the matrix $A$. These vectors are also called the basis vectors explaining the column space of A. For the $i^{th}$ vector in $U$, $u_i$, the amount of the variance explained is proportional to the square of the $i^{th}$ singular value $\sigma_i^2$. Hence the percentage variance explained by a basis vector $u_i$ is given by $\sigma_i^2 / (\sum_{j=1}^{d}(\sigma_j^2))$.

## 4 METHODOLOGY

The pseudocode of our approach is presented in Algorithm 1. Given a forget class, our method aims to suppress the class discriminatory activations from input activations ($a_i$) of that class. When provided with a class-discriminatory projection matrix $P_{dis}$, removing the class discriminatory information from inputs is equivalent to projecting the inputs onto the matrix $I - P_{dis}$. Consider a linear layer $a_o = a_i(\theta^l)^T$, where $\theta^l$ are the weights of the linear layer and $a_o$ is the output activation. Post multiplying the weight with $(I - P_{dis})^T$, is mathematically the same as removing class-discriminatory information from $a_i$. This mechanism allows us to update the model weights to destroy the class-discriminatory activations given the matrix $P_{dis}$ and is done by the *update_parameter()* function in line 15 of Algorithm 1. The rest of the section focuses on optimally computing this class-discriminatory projection matrix. We start with identifying the critical activation space for retain class samples and the forget class samples. These spaces are referred to as the Retain Space ($U_r$) and the Forget Space ($U_f$) respectively and are computed using the SVD on the activations of the corresponding samples. This corresponds to lines 3-7 of Algorithm 1 and the details are presented in Subsection 4.1. In Subsection 4.2 we explain the details for computing $P_{dis}$ and end this section with and discussion on hyperparameter search in Subsection 4.3.

### 4.1 SPACE ESTIMATION

To estimate the Retain Space ($U_r$), we utilize a small subset of samples from the classes to be retained, denoted as $X_r = \{x_i\}_{i=1}^{K_r}$, where $K_r$ represents the number of retain samples. We accumulate a representation matrix, $R_r^l$, in a list $R_r = [R_r^l]_{l=1}^{L}$ for both linear and convolutional layers, where $l$ is layer. In the case of linear layer, this matrix is given by $R_r^l = [(transpose(a_i^l))_{i=1}^{K_r}]$ which is the transpose of the input activation matrix. A convolutional layer has to be represented as a matrix multiplication to apply the proposed weight update rule. This is done using the unfold (Liu et al., 2018) operation on input activations. For a convolutional layer with $C_i$ input channels and $k$ as kernel size each sliding window has the size of $C_i \times k \times k$. If the output activation $a_o$ has the resolution of $h_o \times w_o$, where $h_o$ and $w_o$ is the height and width of the output activation, then

---

**Algorithm 1** Propose SVD based Training Free Algorithm for Controlled forgetting

---

**Input:** $\theta$ is the parameters of the original model; $X_r$ and $X_f$ is a set of few input samples $x_i$ in the retain and forget partition of the train dataset respectively; $\mathcal{D}_{train\_sub\_r}$ and $\mathcal{D}_{train\_sub\_f}$ is the subset of the retain and forget partition of the train dataset; and alpha_r_list and alpha_f_list are list of hyperparameters $\alpha_r$ and $\alpha_f$ respectively.

1. procedure **Unlearn**( $\theta$, $X_r$, $X_f$, $\mathcal{D}_{train\_sub\_r}$, $\mathcal{D}_{train\_sub\_f}$, alpha_r_list, alpha_f_list )
2.     best_score = **get_score**($\theta$, $\mathcal{D}_{train\_sub\_r}$, $\mathcal{D}_{train\_sub\_f}$);     $\theta_f^* = \theta$
3.     $R_r$ = **get_representation**(model, $X_r$)           //Collect representations of linear
4.     $R_f$ = **get_representation**(model, $X_f$)                   and convolution layers
5.     **for** each linear and convolution layer $l$ **do**
6.        $U_r^l, \Sigma_r^l$ = **SVD**($R_r^l$)                 //Retain Space for each layer
7.        $U_f^l, \Sigma_f^l$ = **SVD**($R_f^l$)                 //Forget Space for each layer
8.     **for** each $\alpha_r \in$ alpha_r_list **do**
9.        **for** each $\alpha_f \in$ alpha_f_list **do**
10.           $\theta_f$ = **copy**($\theta$)
11.          **for** each linear and convolution layer $l$ **do**
12.           $\Lambda_r^l$ = **scale_importance**($\Sigma_r^l, \alpha_r$);     $\Lambda_f^l$ = **scale_importance**($\Sigma_f^l, \alpha_f$)     //Eqn. 1
13.           $P_r^l = U_r^l \Lambda_r^l$**transpose**($U_r^l$);     $P_f^l = U_f^l \Lambda_f^l$**transpose**($U_f^l$)
14.           $P_{dis}^l = P_f^l(I - P_r^l)$                 //class discriminatory projection matrix
15.           $\theta_f^l$ =**update_parameter**($I - P_{dis}^l, \theta^l$ ) //orthogonal parameter projection for $l^{th}$ layer
16.          score = **get_score**($\theta_f$, $\mathcal{D}_{train\_sub\_r}$, $\mathcal{D}_{train\_sub\_f}$)
17.          **if** score > best_score **do**
18.             best_score = score;     $\theta_f^* = \theta_f$
19. **return** $\theta_f^*$

---

there are $h_o w_o$ patches of size $C_i \times k \times k$ in the activation $a_i$. The convolution kernel operates on each of these patches in a sliding window fashion to get the values at the corresponding locations in the output map. The unfold operation flattens each of these $h_o w_o$ patches to get a matrix of size $h_o w_o \times C_i kk$. Now, if we reshape the weight as $C_i kk \times C_o$ where $C_o$ is the output channels, we see that the convolution operation becomes a matrix multiplication between the unfolded matrix and the reshaped weights achieving the intended objective. Note that the unfold operation generates $h_o w_o$ samples for each input activation due to the weight sharing property of convolutions and hence we subsample the patches using *subsample()* operation. The representation matrix for the convolutional layer is given by $R_r^l = [(transpose(subsample(unfold(a_i^l))))_{i=1}^{K_r}]$. This explains the *get_representation()* function used in lines 3 and 4 of the Algorithm 1 We perform the SVD on these representation matrices for each layer as shown in line 6 of the Algorithm 1. SVD returns the basis vectors $U_r^l$ that span the activation of the retain samples in $X_r$ and the singular values $\Sigma_r^l$ for each layer $l$. The Retain Space $U_r = [U_r^l]_{l=1}^L$ is the list of these basis vectors for all the layers. Our approach makes a single pass over the samples in $X_r$ to obtain the Retain Space. Forget Space ($U_f$) is estimated similarly on the samples from the class to be unlearnt, denoted by $X_f = \{x_i\}_{i=1}^{K_f}$ where $K_f$ represents the number of samples. We compute the discriminatory projection matrix $P_{dis}$ in the next Subsection by removing the features from the Forget Space that are shared with the Retain Space. This corresponds to lines 12-14 in Algorithm 1. In the next Subsection, we discuss how we isolate the class discriminatory Space.

## 4.2 CLASS-DISCRIMINATORY SPACE

Computing $P_{dis}$ requires evaluating the retain projection matrix $P_r$ and the forget projection matrix $P_f$ using the Retain Space and Forget Space. As the basis vectors in these Spaces are orthonormal and do not capture any information about the amount of input explained by a basis vector. The information of the significance of the basis vector is given by the corresponding singular value. We propose to scale the basis vector in proportion to the amount of variance they explain in the input space as presented below.

**Importance-base Space Scaling ($\Lambda$):** To capture the importance the $i^{th}$ basis vector in the matrix $U$ (or the $i^{th}$ column of $U$), we formulate an diagonal importance matrix $\Lambda$ having the $i^{th}$ diagonal component $\lambda_i$ given by Equation 1. Here $\sigma_i$ represents the the $i^{th}$ singular value in the matrix $\Sigma$. The parameter $\alpha \in (0, \infty)$ called the scaling coefficient is a hyperparameter that controls the scaling of the basis vectors. When $\alpha$ is set to 1 the basis vectors are scaled by the amount of variance

they explain. As $\alpha$ increases the importance score for each basis vector increases and reaches 1 as $\alpha \to \infty$. Similarly, decrease in $\alpha$ decreases the importance of the basis vector and goes to 0 as $\alpha \to 0$. This operation is represented by *scaled_importance()* function in line 12 of the Algorithm 1. It is important to note that without the proposed scaling approach the matrices $P_r$ and $P_f$ become identity in line 13 of Algorithm 1, as $U$ is an orthonormal matrix. This inturn makes $P_{dis}$ a zero matrix, which means the weight update in line 15 of Algorithm 1 projects weight on identity matrix mathematically restricting unlearning. Hence it is important to use scaling in Line 12 of Algorithm 1.

$$\lambda_i = \frac{\alpha \sigma_i^2}{(\alpha - 1)\sigma_i^2 + \sum_{j=1}^{d} \sigma_j^2}, \text{ where d is the number of basis vectors.} \tag{1}$$

**Class Discriminatory Projection Matrix**($P_{dis}$)**:** Say we have Spaces $U_r^l$ and $U_f^l$ for a layer and the scaling coefficients are set to a value of $\alpha_r$ and $\alpha_f$. We can compute the importance scaling matrices $\Lambda_r$ and $\Lambda_f$ as per Equation1. The retain projection matrix, which projects the input activations to the retain space is given by $P_r^l = U_r^l \Lambda_r^l (U_r^l)^T$ and the forget projection matrix given by $P_f^l = U_f^l \Lambda_f^l (U_f^l)^T$, see line 13 in Algorithm 1. To obtain the unlearn class discriminatory projection matrix, we remove the shared space given by $P_f^l P_r^l$ from the forget projection matrix to obtain $P_{dis}^l = P_f^l - P_f^l P_r^l$. Alternatively, this can also be written as $P_{dis}^l = P_f^l(I - P_r^l)$ which projects the forget projection matrix on the orthogonal retain space. Intuitively, this projects the forget space onto the space that does not contain any information about the retain space ( or orthogonal retain space), effectively removing the shared information from the forget projection matrix to obtain the discriminatory projection space. The parameters of the convolutional layer needs to be reshaped to $C_i kk \times C_o$ before being projected on $(I - P_{dis}^l)$. Our algorithm introduces two hyperparameters namely $\alpha_r$ and $\alpha_f$ the scaling coefficients for the Retain Space and the Forget Space respectively. The next Subsection 4.3 presents a discussion on these hyperparameters.

### 4.3 HYPERPARAMETER SEARCH

Our algorithm searches for the optimal hyperparameter values for $\alpha_r$ and $\alpha_f$ within the predefined lists provided as *alpha_r_list* and *alpha_f_list*, respectively. We observe this search is necessary for our algorithm. One intuitive explanation for this is that the unlearning class may exhibit varying degrees of confusion with the retain classes, making it easier to unlearn some classes compared to others, hence requiring different scaling for the retain and forget spaces. We introduce a simple scoring function *get_score()* which assesses the quality of the unlearnt model for a given pair of $\alpha_r$ and $\alpha_f$. The *get_score()* function returns penalized retain accuracy given by $score = acc_r(1 - acc_f/100)$, where $acc_r$ and $acc_f$ are the accuracy on the $\mathcal{D}_{train\_sub\_r}$ and $\mathcal{D}_{train\_sub\_f}$ respectively. Our algorithm begins with the best unlearn model parameters $\theta_f^*$ as the original model parameters $\theta$. Finally as seen in line 8 and 9 of the Algorithm 1 we do a grid search over all the possible values of $\alpha_r$ and $\alpha_f$ provided in *alpha_r_list* and *alpha_f_list* to obtain the best unlearnt parameters $\theta_f^*$. Note, we observe that increasing the value of $\alpha_f$ decreases the retain accuracy $acc_r$ and hence we terminate the inner loop (line 9) to speed up the grid search and have not presented this in the Algorithm 1 for simplicity.

**Discussion:** The speed and efficiency of our approach can be attributed to the design choices. Firstly our method runs inference for very few samples to get the representations $R$. Further, the small sizes of these representation matrices ensure that SVD is fast and computationally cheap. Additionally, the SVD operation for each layer is independent and can be parallelized to further improve speeds. Our approach only performs inference and does not rely on computationally intensive gradient based optimization steps (which also require tuning the learning rates) and gets the unlearnt model in a single step for each grid search (over $\alpha_r$ and $\alpha_f$) leading to a fast and efficient approach. Additionally, our method has fewer parameters as compared to the gradient based baselines which are sensitive to the choices of optimizer, learning rate, batch size, learning rate scheduler, weight decay, etc. Further, our algorithm can be readily extended to Transformer architectures by applying our algorithm to all the linear layers in the architecture. Note, that we do not change the normalization layers for any architecture as the fraction of total parameters for these layers is insignificant. Further, our algorithm assumes a significant difference in the distribution of forget and retain samples for SVD to find distinguishable spaces. This is true in class unlearning setup, where retain and forget samples come from different non-overlapping classes. However, in the case of unlearning a random subset of training data, this assumption would not hold and our method has limited performance in such a scenario, requiring additional modification for effective unlearning.

## 5 EXPERIMENTS

**Dataset and Models :** We conduct the class unlearning experiments on CIFAR10, CIFAR100 (Krizhevsky et al., 2009) and ImageNet (Deng et al., 2009) datasets. We use the modified versions of ResNet18 (He et al., 2016) and VGG11 (Simonyan & Zisserman, 2014) with batch normalization for the CIFAR10 and CIFAR100 datasets. These models are trained for 350 epochs using Stochastic Gradient Descent (SGD) with the learning rate of 0.01. We use Nesterov (Sutskever et al., 2013) accelerated momentum with a value of 0.9 and the weight decay is set to 5e-4. For the ImageNet dataset, we use the pretrained VGG11 and base version of Vision Transformer with a patch size of 14 (Dosovitskiy et al., 2020) available in the torchvision library.

**Comparisons:** We benchmark our method against 5 unlearning approaches. Two of these approaches, *Retraining* (Chundawat et al., 2023) and *NegGrad* (Tarun et al., 2023) are commonly used in literature. Retraining involves training the model from scratch using the retain partition of the training set, $D_{train\_r}$, and serves as our gold-standard model. In the NegGrad approach, we finetune the model for a few step using gradient ascent on the forget partition of the train set $D_{train\_f}$ with a gradient clipping threshold set at 0.25. This approach ensures good forgetting, however is seen to reduce the model accuracy on the retain partition. We also compare our approach to a stronger version of NegGrad called *NegGrad+* proposed by Kurmanji et al. (2023). This algorithm does gradient ascent on the forget samples and gradient descent on the retain samples for 500 steps. A detailed explanation of NegGrad and NegGrad+ with psuedocodes is presented in Appendix A.2 and A.3 respectively. Finally, we compare our work with two SoTA algorithms (Tarun et al., 2023; Kurmanji et al., 2023) to demonstrate the effectiveness of our approach. Discussion on hyperparameters is presented in Appendix A.4.

**Evaluation:** Our experiments evaluate the accuracy on the unlearnt models with the accuracy on retain samples $ACC_r$ and the accuracy on the forget samples $ACC_f$. In addition, we implement Membership Inference Attack ($MIA$) to distinguish between samples in $\mathcal{D}_{train\_r}$ and $\mathcal{D}_{test\_r}$. We use the confidence scores for the target class and training a Support Vector Machine (SVM) (Hearst et al., 1998) classifier. We report the average model predictions on $\mathcal{D}_{train\_f}$ as the MIA scores in our evaluations. A high value of $MIA$ score for a given sample indicates that it does not belong to the training data. An unlearnt model is expected to match the $MIA$ score of the Retrained model. See Appendix A.10 for details on MIA experiment.

## 6 RESULTS

**Table 1:** Results for Single class Forgetting on CIFAR10 and CIFAR100 dataset. (We bold font the row having highest value for $ACC_r(100 - ACC_f)MIA$)

| | Method | VGG11_BN | | | ResNet18 | | |
|---|---|---|---|---|---|---|---|
| | | $ACC_r(\uparrow)$ | $ACC_f(\downarrow)$ | $MIA(\uparrow)$ | $ACC_r(\uparrow)$ | $ACC_f(\downarrow)$ | $MIA(\uparrow)$ |
| CIFAR10 | Original | $91.58 \pm 0.52$ | $91.58 \pm 4.72$ | $0.11 \pm 0.08$ | $94.89 \pm 0.31$ | $94.89 \pm 2.75$ | $0.03 \pm 0.03$ |
| | Retraining | $92.58 \pm 0.83$ | $0$ | $100 \pm 0$ | $94.81 \pm 0.52$ | $0$ | $100 \pm 0$ |
| | NegGrad | $81.46 \pm 5.67$ | $0.02 \pm 0.04$ | $0$ | $69.89 \pm 10.23$ | $0$ | $0$ |
| | NegGrad+ | $89.79 \pm 1.49$ | $0.13 \pm 0.16$ | $99.93 \pm 0.15$ | $87.38 \pm 1.36$ | $0.2 \pm 0.33$ | $0$ |
| | Tarun et al. (2023) | $89.21 \pm 0.84$ | $0$ | $0$ | $92.20 \pm 0.72$ | $10.89 \pm 8.79$ | $61.5 \pm 25.86$ |
| | Kurmanji et al. (2023) | $46.18 \pm 35.376$ | $0$ | $0$ | $80.28 \pm 7.31$ | $6.4 \pm 19.074$ | $0$ |
| | Ours | $\mathbf{91.77 \pm 0.69}$ | $\mathbf{0}$ | $\mathbf{98.28 \pm 5.43}$ | $\mathbf{94.19 \pm 0.50}$ | $\mathbf{0.03 \pm 0.09}$ | $\mathbf{95.5 \pm 14.23}$ |
| CIFAR100 | Original | $69.22 \pm 0.29$ | $67 \pm 15.23$ | $0.2 \pm 0.25$ | $76.64 \pm 0.13$ | $74.3 \pm 13.27$ | $0.08 \pm 0.1$ |
| | Retraining | $68.97 \pm 0.40$ | $0$ | $100 \pm 0$ | $76.81 \pm 0.50$ | $0$ | $100 \pm 0$ |
| | NegGrad | $51.21 \pm 6.37$ | $0$ | $0$ | $60.32 \pm 7.03$ | $0$ | $29.98 \pm 48.27$ |
| | NegGrad+ | $58.66 \pm 3.91$ | $0$ | $0$ | $71.37 \pm 2.78$ | $0$ | $100 \pm 0$ |
| | Tarun et al. (2023) | $53.94 \pm 1.22$ | $0$ | $0$ | $63.387 \pm 0.50$ | $3.1 \pm 5.65$ | $0$ |
| | Kurmanji et al. (2023) | $67.073 \pm 0.41$ | $10.5 \pm 18.32$ | $84.3 \pm 26.76$ | $72.54 \pm 0.43$ | $10.2 \pm 16.90$ | $89.28 \pm 17.18$ |
| | Ours | $\mathbf{65.94 \pm 1.21}$ | $\mathbf{0.3 \pm 0.48}$ | $\mathbf{99.92 \pm 0.10}$ | $\mathbf{73.60 \pm 1.41}$ | $\mathbf{0.3 \pm 0.48}$ | $\mathbf{100 \pm 0}$ |

**Table 2:** Results for Single class forgetting on ImageNet-1k dataset.

| Method | Total samples | VGG11_BN | | | ViT_B_16 | | |
|---|---|---|---|---|---|---|---|
| | | $ACC_r(\uparrow)$ | $ACC_f(\downarrow)$ | $MIA(\uparrow)$ | $ACC_r(\uparrow)$ | $ACC_f(\downarrow)$ | $MIA(\uparrow)$ |
| Original | - | $68.61 \pm 0.02$ | $72.6 \pm 25.92$ | $22.72 \pm 22.59$ | $80.01 \pm 0.037$ | $80.6 \pm 19.87$ | $13.36 \pm 12.94$ |
| NegGrad+ | 32000 | $66.37 \pm 1.27$ | $8.8 \pm 11.48$ | $96.58 \pm 4.40$ | $73.76 \pm 1.46$ | $0$ | $99.98 \pm 0.05$ |
| Tarun et al. (2023) | 9990 | $43.5618 \pm 0.59$ | $0$ | $98.96 \pm 3.26$ | $56.00 \pm 3.47$ | $38.8 \pm 34.074$ | $66.67 \pm 50$ |
| Kurmanji et al. (2023) | 10000 | $\mathbf{67.29 \pm 0.34}$ | $\mathbf{0}$ | $\mathbf{99.92 \pm 0.15}$ | $79.23 \pm 0.19$ | $56 \pm 21.56$ | $45.47 \pm 20.62$ |
| Ours | 1499 | $66.41 \pm 0.60$ | $0.6 \pm 1.35$ | $99.33 \pm 0.90$ | $\mathbf{78.47 \pm 0.84}$ | $\mathbf{0.2 \pm 0.63}$ | $\mathbf{99.98 \pm 0.05}$ |

**Class Forgetting:** We present the results for single class forgetting in Table 1 for the CIFAR10 and CIFAR100 dataset. The table presents results that include both the mean and standard deviation across 10 different target unlearning classes. CIFAR10 dataset is accessed for unlearning on each class and CIFAR100 is evaluated for every 10th starting from the first class. The Retraining

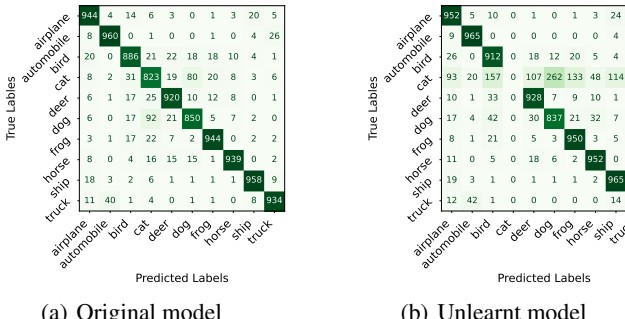

(a) Original model                    (b) Unlearnt model

**Figure 2:** Confusion Matrix for the original VGG11 model and a model unlearning cat class using our algorithm, showing redistribution of cat samples to other classes in proportion to the confusion in original model.

approach matches the accuracy of the original model on retain samples and has $0\%$ accuracy on the forget samples, which is the expected upper bound. The $MIA$ accuracy for this model is $100\%$ which signifies that MIA model is certain that $\mathcal{D}_{train\_f}$ does not belong to the training data. The NegGrad method shows good forgetting with low $ACC_f$, however, performs poorly on $ACC_r$ and $MIA$ metrics. The NegGrad algorithm's performance on retain samples is expected to be poor because it lacks information about the retain samples required to protect the relevant features. Further, this explains why NegGrad+ which performs gradient descent on the retain samples along with Neg-Grad can maintain impressive performance on retain accuracy with competitive forget accuracy. In some of our experiments, we observe that the NegGrad+ approach outperforms the SoTA benchmarks (Tarun et al., 2023) and (Kurmanji et al., 2023) which suggests the NegGrad+ approach is a strong baseline for class unlearning. Our proposed training free algorithm achieves a better tradeoff between the evaluation metrics when compared against all the baselines. Further, we observe the MIA numbers for our method close to the retrained model and better than all the baselines for most of our experiments. We demonstrate our algorithm easily scales to ImageNet without compromising its effectiveness, as seen in Table 2. Due to the training complexity of the experiments, we are not able to obtain retrained models for ImageNet. We observe that the results on CIFAR10 and CIFAR100 datasets consistently show $ACC_f$ to be 0 and the $MIA$ performance to be $100\%$. We, therefore, interpret the model with high $ACC_r$, $MIA$, and low $ACC_f$ as a better unlearnt model for these experiments. We conduct unlearning experiments on the ImageNet dataset for every 100th class starting from the first class, resulting in a total of 10 experiments. Our algorithm shows less than $1.5\%$ drop in $ACC_r$ as compared to the original model while maintaining less than $1\%$ forget accuracy for a well trained SoTA Transformed based model. The MIA scores for our model are nearly $100\%$ indicating that model the MIA model fails to recognize $\mathcal{D}_{train\_f}$ as part of training data. We observe that (Kurmanji et al., 2023) outperforms our method for a VGG11 model trained on ImageNet, however, requires access to $6\times$ more samples and requires more compute than our approach.

**Confusion Matrix analysis:** We plot the confusion matrix showing the distribution of true labels and predicted labels for the original VGG11 model and VGG11 model unlearning cat class with our algorithm for CIFAR10 in Figure 2. Interestingly, we observe that a significant portion of the cat samples are redistributed across the animal categories. The majority of these samples are assigned to the dog class, which exhibited the highest level of confusion with the cat class in the original

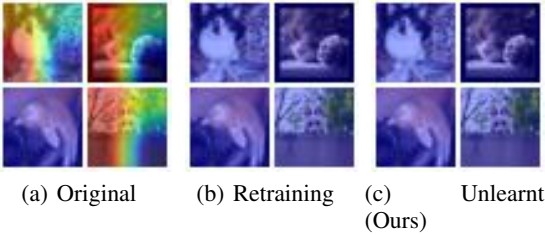

(a) Original     (b) Retraining     (c)      Unlearnt
                                    (Ours)

**Figure 3:** GradCAM-based heatmaps for (a) Original, (b) Retrained, and (c) Unlearnt VGG11 model on the CIFAR10 with a cat as the target class, demonstrating that the unlearned model does not highlight any features specific to the cat.

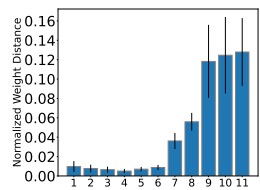

**Figure 4:** Layer-wise weight change for VGG11 on cifar10 dataset.

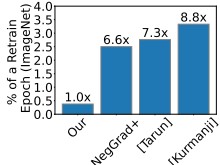

**Figure 5:** Compute comparison for single linear layer of ViT.

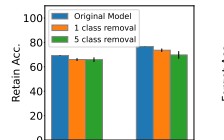 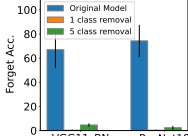 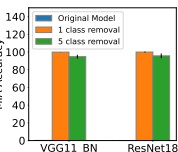

(a) Retain Accuracy. (b) Forget Accuracy (c) MIA accuracy

**Figure 6:** Multi-Class Unlearning for CIFAR100 dataset.

model. This aligns with the illustration shown in Figure 1 where the forget space gets redistributed to the classes in the proximity of the forget class. We present the confusion matrix of the retrained VGG11 model without the cat class in Figure 7 in Appendix A.5. This matrix also shows a high number of cat samples being assigned to the dog class.

**Saliency-based analysis :** We test this VGG11 model unlearning the cat class with GradCAM-based feature analysis as presented in Figure 3, and we see that our model is unable to detect class discriminatory information. This behavior is similar to the retrained VGG11 model shown in Figure 3(b).

**Layer-wise analysis:** We plot the change layer-wise weight difference between the parameters of the unlearnt and the original model for VGG11 on the cifar10 dataset in Figure 4. We observe that the weight change is larger for the later layers. This is expected as the later layers are expected to learn complex class discriminatory information while the initial layers do learn edges and simple textures (Olah et al., 2017). We present the retain and forget accuracy when our algorithm is applied to the top layers in Appendix A.6.

**Compute analysis:** We analytically calculate the computational cost for different unlearning algorithms for a Vision Transformer (ViT) model trained on ImageNet, as illustrated in Figure 5, see Appendix A.7 for details. This figure shows the percentage of compute cost as compared to a single epoch of retraining baseline on y axis. It's important to note that we exclusively consider the computation of the linear layer ignoring the compute costs for self attention and normalization layers. This inherently works in favor of the gradient based approaches as our algorithm has significantly low overhead for these layers as we only do forward pass on a few samples while representation collection. Our approach demonstrates more than $6\times$ compute reduction than any other baseline.

**Multi Class Forgetting:** The objective of Multiclass removal is to remove more than one class from the trained model. In multi task learning a deep learning model is trained to do multiple tasks where each of the tasks is a group of classes. The scenario of One-Shot Multi-Class where the unlearning algorithm is expected to remove multiple classes in a single unlearning step has a practical use case in such task unlearning. Our algorithm estimates the Retain Space $U_r$ and the Forget Space $U_f$ based on the samples from $X_r$ and $X_f$. It is straightforward to scale our approach to such a scenario by simply changing the retain sample $X_r$ and $X_f$ to represent the samples from class to be retained and forgotten respectively. We demonstrate multi class unlearning on removing 5 classes belonging to a superclass on CIFAR100 dataset in Figure 6. We observe our method is able to retain good accuracy on Retain samples and has above $95\%$ MIA accuracy while maintaining a low accuracy on forget set under this scenario. When compared with Tarun et al. (2023) under this unlearning setting (see Table 6 in Appendix A.8) we see our method has significantly better performance. We also present results of multiclass unlearning on CIFAR10 in Appendix A.8. Additionally, we present a sequential version of multi class unlearning in Appendix A.9.

## 7 CONCLUSION

In this work, we introduce a novel class and multi-class unlearning algorithm based on Singular Value Decomposition (SVD), which eliminates the need for gradient-based unlearning steps. We demonstrate the efficacy of our approach over a variety of image classification datasets and network architectures. Our algorithm consistently performs better than SoTA on several evaluation metrics while being much more computationally efficient. Furthermore, to the best of our knowledge, our proposed class unlearning algorithm is the first to be demonstrated on large-scale datasets like ImageNet with a SoTA transformer based model. Our analysis, conducted through saliency-based explanations, does not reveal the class-discriminatory features, and the confusion matrix analysis shows the redistribution of the unlearned samples based on their confusion with respective classes.

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

# A APPENDIX

## A.1 DEMONSTRATION WITH TOY EXAMPLE

In Figure 1 we demonstrate our unlearning algorithm on a 4 way classification problem, where the original model is trained to detect samples from 4 different 2-dimensional Gaussian's centered around (1,1), (-1,1), (-1, -1) and (1, -1) respectively with a standard of (0.5,0.5). The training dataset has 10000 samples per class and the test dataset has 1000 samples per class. The test data is shown with dark points in the decision boundaries in Figure 1. We use a simple 5-layer linear model with ReLU activation functions. All the intermediate layers have 5 neurons and each layer excluding the final layer is followed by BatchNorm. We train this network with stochastic gradient descent for 10 epochs with a learning rate of 0.1 and Nestrove momentum of 0.9. The decision boundary learnt by the original model trained on complete data is shown in Figure 1(a) and the accuracy of this model on test data is 95.60%. In Figure 1(b) we plot the decision boundary for the model obtained by unlearning the class with mean (1,1) with our algorithm. This decision boundary is observed to be close to the decision boundary of the model retrained without the data points from class with mean (1,1) as shown in Figure 1(c). The accuracy of the unlearnt model is 97.43% and retrained model is 97.33%. This illustration shows that the proposed algorithm redistributes the input space of the class to be unlearnt to the closest classes.

## A.2 NEGGRAD ALGORITHM

Pseudocode for NegGrad is presented in Algorithm 2. The algorithm initialized the unlearn parameters $\theta_f^*$ to the original parameters $\theta$ and does 500 steps of gradient ascent on the forget subset of the training data. After every 100 steps, we evaluate the model accuracy on $\mathcal{D}_{train\_sub\_f}$ and exit ascent when $acc_f$ becomes lower than 0.1. This restricts the gradient ascent from catastrophically forgetting the samples in the retain partition.

---

**Algorithm 2** NegGrad Algorithm

**Input:** $\theta$ is the parameters of the original model, $\mathcal{L}$ is the loss function, $\mathcal{D}_{train\_sub\_f}$ is the subset of the forget partition of the train dataset; and $\eta$ is the learning rate

1. procedure **Unlearn**( $\theta$, $\mathcal{L}$, $\mathcal{D}_{train\_sub\_f}$, $\eta$ )
2.     $\theta_f^* = \theta$
3.     **for** step = 1,...., 500 **do**
4.         input, target = **get_batch**($\mathcal{D}_{train\_sub\_f}$)
5.         $g$ = **get_gradients**($\theta_f^* f$, $\mathcal{L}$, input, target)
6.         $g$ = **gradient_clip**($g$,0.25)
7.         $\theta_f^* = \theta_f^* + \eta g$
8.         **if** step multiple of 100 **do**
9.             $acc_f$ = **get_accuracy**($\theta$, $\mathcal{D}_{train\_sub\_f}$)
10.             **breakif** $acc_f < 0.1$
11. **return** $\theta_f^*$

---

## A.3 NEGGRAD+ ALGORITHM

NegGrad+ is a superior gradient ascent unlearning algorithm as compared to the NegGrad. Algorithm 3 outlines the pseudocode for the NegGrad+ unlearning approach. The algorithm initializes the unlearn parameters $\theta_f^*$ to the original parameters $\theta$ and gets the model accuracy on the forget partition $acc_f$. The gradients $g_a$ are computed on the forget partition if the $acc_f$ is greater than 0.1 otherwise $g_a$ is set to 0. The gradient on the retain batch denoted by $g_d$ is computed at every step and the unlearn parameters are updated in the descent direction for the retain samples and ascent direction for the forget samples. The values of $acc_f$ is updated after every 100 steps. This algorithm mitigates the adverse effect of Naive descent on the retain accuracy. Once the model achieves the forget accuracy less than 0.1 the algorithm tries to recover the retain accuracy by finetuning on the retain samples.

---

**Algorithm 3** NegGrad+ Algorithm

---

**Input:** $\theta$ is the parameters of the original model; $\mathcal{L}$ is the loss function; $\mathcal{D}_{train\_sub\_f}$ and $\mathcal{D}_{train\_sub\_f}$ are the subset of the retain and forget partition of the train dataset respectively; and $\eta$ is the learning rate.

1. procedure **Unlearn**( $\theta$, $\mathcal{L}$, $\mathcal{D}_{train\_sub\_r}$, $\mathcal{D}_{train\_sub\_f}$, $\eta$ )
2.     $acc_f$ = **get_accuracy**($\theta$, $\mathcal{D}_{train\_sub\_f}$);     $\theta_f^* = \theta$
3.     **for** step = 1,...., 500 **do**
4.         **if** $acc_f > 0.1$ **do**
5.             input, target = **get_batch**($\mathcal{D}_{train\_sub\_f}$)
6.             $g_a$ = **get_gradients**($\theta_f^* f$, $\mathcal{L}$, input, target)
7.             $g_a$ = **gradient_clip**($g_a$,0.25)
8.         **else**
9.             $g_a = 0$
10.         input, target = **get_batch**($\mathcal{D}_{train\_sub\_r}$)
11.         $g_d$ = **get_gradients**($\theta_f^* f$, $\mathcal{L}$, input, target)
12.         $\theta_f^* = \theta_f^* + \eta g_a - \eta g_d$
13.         **if** step multiple of 100 **do**
14.             $acc_f$ = **get_accuracy**($\theta$, $\mathcal{D}_{train\_sub\_f}$)
15. **return** $\theta_f^*$

---

### A.4 HYPERPARAMETER DISCUSSION

Our approach introduces four key hyperparameters: the list of $\alpha_r$ values (alpha_r_list), the list of $\alpha_f$ values (alpha_f_list), and the number of samples used to estimate the Retain Space and Forget Space. The values for these hyperparameters are dependent on the dataset and are presented in Table 3 of Appendix A.4. The NegGrad and NegGrad+ require tuning of the learning rate $\eta$ for atleast 1 unlearning class and a list of the learning rates is presented in Table 4 in Appendix A.4. We tune this hyperparameter for unlearning the first class on each model-dataset pair. Once determined, these hyperparameters remain fixed for unlearning all other classes. The SoTA (Tarun et al., 2023) baseline introduces 2 learning rates for the impair and the repair stages represented by $\eta_{impair}$ and $\eta_{repair}$. Similar to the other baselines these hyperparameters are only tuned on one class for each model-dataset pair. For (Kurmanji et al., 2023) we use all the suggested hyperparameters given in the work for Large scale experiments on CIFAR10 for class unlearning-type (Table 3) and tune the batch sizes (forget-set bs and retain-set bs) as given in Table 5 The Retraining method does not add any additional hyperparameters and is trained with the same hyperparameters as the original model.

**Table 3:** Hyperparameters for our approach with single class unlearning or sequential multi-class unlearning.

| Dataset | alpha_r_list | alpha_f_list | samples/class in $X_r$ | samples/class class in $X_f$ |
|---|---|---|---|---|
| CIFAR10 | [10, 30, 100, 300, 100] | [3] | 100 | 900 |
| CIFAR100 | [100, 300, 1000] | [3, 10, 30, 100] | 10 | 990 |
| ImageNet | [30, 100, 300, 1000, 3000] | [3, 10, 30, 100, 300] | 1 | 500 |

**Table 4:** Hyperparameter tuning space for NegGrad, NegGrad+ and (Tarun et al., 2023) benchmarks.

| Method | $\eta$ **or** $\eta_{repair}$ **or** $\eta_{impair}$ |
|---|---|
| NegGrad | [1e-4, 2e-4,5e-4,1e-3,2e-3,5e-3,1e-2] |
| NegGrad+ | [1e-4, 2e-4,5e-4,1e-3,2e-3,5e-3,1e-2] |
| Tarun et al. (2023) | [1e-4, 2e-4,5e-4,1e-3,2e-3,5e-3,1e-2] |

**Table 5:** Hyperparameters for our approach with single class unlearning or sequential multi-class unlearning.

| Dataset | forget-set batch size | retain-set batch size |
|---|---|---|
| CIFAR10 and CIFAR100 | [32, 64, 128, 256, 512] | [32, 64, 128, 256, 512] |
| ImageNet | [32, 64, 128, 256] | [32, 64, 128, 256] |

A.5 CONFUSION MATRIX FOR RETRAINED MODEL

Figure 7 shows the confusion matrix for the VGG11 model retrained without cat class.

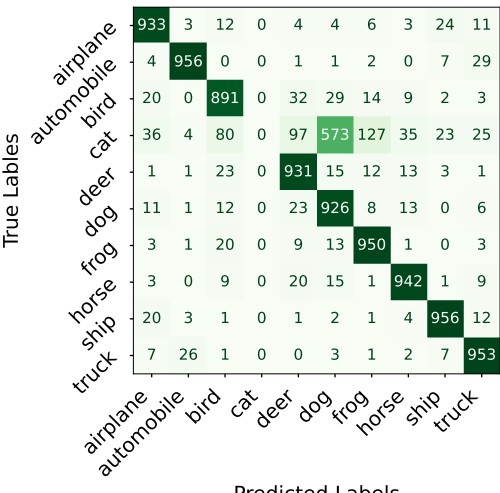

Predicted Labels

**Figure 7:** Confusion Matrix retrained VGG11 model without the cat class sample on VGG11 model with batchnorm.

A.6 EFFECT OF APPLYING OUR ALGORITHM TO DIFFERENT LAYERS

Figure 8 shows the results for the unlearning model obtained when our algorithm is applied only to the top layers starting from $n^{th}$ layer to the end of the network for CIFAR10 dataset on VGG11 model. The x-axis represents the number of initial layers $n$ we do not apply unlearning algorithm in the plot. When the value of $n = 0$ our algorithm is applied to the entire model for all the other values of $n$ on x-axis of Figure 8 represents a case where we do not change the initial layers $0 - n$(including $n$) in unlearning. We observe that the effect of removing the projections is minimal on the Retain accuracy. The forget accuracy keeps increasing as we sequentially remove the projections starting from the initial layers. This is the expected trend as the class discriminatory information is expected to be concentrated towards the later layers.

A.7 COMPUTE ANALYSIS FOR SINGLE LAYER OF VIT ON IMAGENET DATASET.

A.7.1 LINEAR LAYER COMPUTE EQUATIONS

Here we analyze the compute required for a linear layer. Say we have a linear layer of size $f_{in} \times f_{out}$, where $f_{in}$ is the input features and $f_{out}$ is the output features. Let the retain set have $n_r$ samples. The input activation for this layer will hence be of size $n_r \times f_{in}$. Below we analyze the compute required by various algorithms in this setting. We substitute all the parameters in the equation to obtain the compute in terms of $f_{in}$ and $f_{out}$.

**Retraining:** The compute required for this method will be $n_r f_{in} f_{out}$ for forward pass and $2n_r f_{in} f_{out}$ for backward resulting in total compute given by Equation 2. For the ImageNet experiments, $n_r$ is approximately 128000. Note this is the compute for the single epoch.

$$C_{retrain}^{\text{Linear}}(f_{in}, f_{out}) = 3n_r f_{in} f_{out} = 3840000 f_{in} f_{out} \qquad (2)$$

**NegGrad/NegGrad+:** The NegGrad and NegGrad+ algorithm make $s_{ng}$ ascent/descent steps with a batchsize of $b_{ng}$. The compute for this would be given by Equation 3. For ImageNet runs on Vit $s_{ng} = 500$ and $b_{ng} = 64$.

$$C_{neggrad}^{\text{Linear}}(f_{in}, f_{out}) = 3s_{ng} b_{ng} f_{in} f_{out} = 96000 f_{in} f_{out} \qquad (3)$$

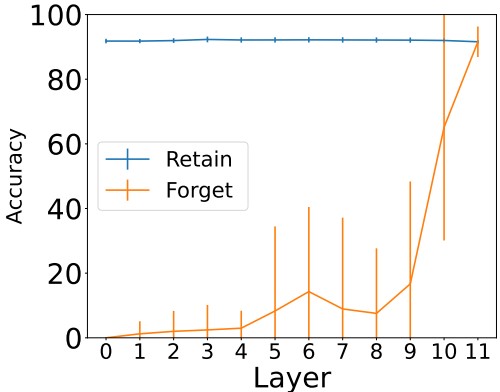

**Figure 8:** Quality of unlearnt model for the unlearning applied to the initial layers for CIFAR10 dataset on VGG11 network.

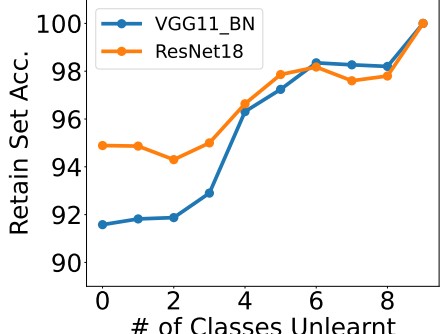

**Figure 9:** Sequential class removal on CIFAR10 dataset.

**(Tarun et al., 2023):** For this baseline the authors generate the noise distribution for each forget class. This is done through gradient ascent on the model for $s_{noise}$ steps starting with a random noise with a batch size of $b_{noise}$. In the impair step, the algorithm performs gradient descent on $n_{impair}$ samples and $n_{Tarun\_r}$ retain samples to remove the forget samples for $s_{impair}$ steps. In the repair steps the model does gradient descent on $n_{Tarun\_r}$ samples to gain performance on retain samples for $s_{repair}$ steps. The compute equation is given by Equation 4 The parameters values are $s_{noise} = 40, b_{noise} = 256, n_{impair} = 5120, s_{impair} = 1, n_{Tarun\_r} = 9990, s_{repair} = 1$.

$$C_{Tarun}^{\text{Linear}}(f_{in}, f_{out}) = 3(\underbrace{s_{noise}b_{noise}}_{\text{Noise Generation}} + \underbrace{s_{impair}(n_{impair} + n_{Tarun\_r})}_{\text{Impair Steps}} + \underbrace{s_{repair}n_{Tarun\_r}}_{\text{Repair Steps}})f_{in}f_{out}$$

$$= 106020 f_{in}f_{out}$$

$$(4)$$

**(Kurmanji et al., 2023):** The author perform $s_{max}$ number of maximization steps on $n_{scrub\_f}$ samples and $s_{min}$ number of maximization steps on $n_{scrub\_r}$. Further, this work uses distillation loss which requires additional forward passes for every step of minimization and maximization. This is given by Equation 5. The values of hyperparameters are $s_{max} = 2, n_{scrub\_f} = 1000, s_{min} = 3, n_{scrub\_r} = 10000$. Note the scaling factor of 4 in the equation accounts for the forward pass in the distillation step.

$$C_{scrub}^{\text{Linear}}(f_{in}, f_{out}) = 4(\underbrace{s_{max}n_{scrub\_f}}_{\text{Maximization Step}} + \underbrace{s_{min}n_{scrub\_r}}_{\text{Minimization Step}})f_{in}f_{out}$$

$$= 128000 f_{in}f_{out}$$

$$(5)$$

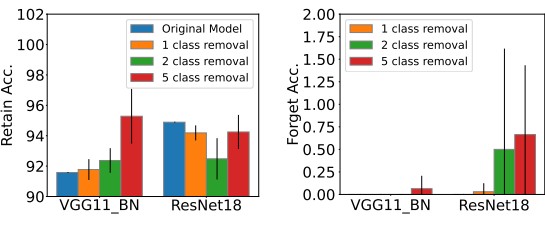

(a) Retain Accuracy.  (b) Forget Accuracy

**Figure 10:** One-Shot Multi-Class Unlearning for CIFAR10 dataset.

**Table 6:** Results for Multi class Forgetting on CIFAR100 dataset.

| Method | VGG11_BN | | | ResNet18 | | |
|---|---|---|---|---|---|---|
| | $ACC_r(\uparrow)$ | $ACC_f(\downarrow)$ | $MIA(\uparrow)$ | $ACC_r(\uparrow)$ | $ACC_f(\downarrow)$ | $MIA(\uparrow)$ |
| NegGrad+ | $57.75 \pm 1.40$ | $0.2 \pm 0.2$ | $0$ | $70.55 \pm 1.045$ | $0.7 \pm 0.12$ | $99.86 \pm 0.11$ |
| Tarun et al. (2023) | $54.31 \pm 1.09$ | $0$ | $0$ | $64.16 \pm 1.18$ | $1.34 \pm 2.33$ | $60.2 \pm 52.14$ |
| Ours | $65.94 \pm 1.89$ | $4.6 \pm 1.44$ | $94.8 \pm 2.2$ | $69.74 \pm 3.12$ | $2.33 \pm 1.61$ | $95.7 \pm 2.67$ |

**Ours:** We have access to $n_{our\_r}$ and $n_{our\_f}$ retain and forget samples in our algorithm and for ImageNet these are $999$ and $500$ respectively. Our approach can be broken into 4 compute steps, namely representation collection, SVD, Space Estimation, and weight projection. Representation collection requires forward pass on a few samples and can be compute cost can be computed as mentioned before. For a matrix of size $m \times n$ SVD has a compute of $mn^2$ Cline & Dhillon (2006) where $m > n$. Space Estimation and weight projection steps involve matrix multiplication. For a matrix A of size $m \times n$ and matrix B of size $n \times p$ the compute costs of matrix multiplication $A \times B$ is $mnp$.

$$C_{our}^{\text{Linear}}(f_{in}, f_{out}) = \underbrace{(n_{our\_r} + n_{our\_f})f_{in}f_{out}}_{\text{Representation Matrix}} + \underbrace{(n_{our\_r} + n_{our\_f})f_{in}^2}_{\text{SVD}} + \underbrace{2f_{in}^3}_{P_{dis} \text{ Computation}}$$
$$+ \underbrace{f_{in}^2 f_{out}}_{\text{Weight Projection}} \tag{6}$$
$$= 1499 f_{in}f_{out} + 1499 f_{in}^2 + 2f_{in}^3 + f_{in}^2 f_{out}$$

### A.7.2 COMPUTE FOR A LAYER OF VIT

A layer of ViT$_{\text{Base}}$ has 4 layers of size 768, namely Key weights, Query weights, value weights, and output weights in the Attention layer. The MLP layer consists of a layer of size $768 \times 3072$ and $3072 \times 768$. The total compute for a layer of Vit would be given by Equation 7. Note this ignores the compute of the attention and normalization layers. Adding compute for the attention mechanism would only benefit our method as we only compute this for representation collection, whereas the baseline methods would have this computation at every forward and backward pass. These equations are used to obtain the numbers for each of the methods in Figure 5.

$$C^{\text{ViT\_Layer}} = 4C^{\text{Linear}}(768, 768) + C^{\text{Linear}}(768, 3072) + C^{\text{Linear}}(3072, 768) \tag{7}$$

### A.8 MULTI CLASS UNLEARNING

We run experiments for this scenario on the CIFAR10 dataset with VGG11 and ResNet18 models. Figure 10 presents the mean and standard deviations for retain accuracy and the forget accuracy for 5 runs on each configuration. The set of classes to be removed is randomly selected for each of these 5 runs. These results show our algorithm scales to this scenario without losing efficacy. Additionally we present the comparisons with baselines on the CIFAR100 dataset for unlearning a superclass in Table 6.

**Table 7:** Location of Projection. Experiments on CIFAR10 dataset similar to Table 1

| Method | VGG11_BN | | ResNet18 | |
| --- | --- | --- | --- | --- |
| | $acc_r$ | $acc_f$ | $acc_r$ | $acc_f$ |
| Original | 91.58 | | 94.89 | |
| input activation suppression (main paper) | $91.77 \pm 0.69$ | 0 | $94.19 \pm 0.50$ | $0.03 \pm 0.09$ |
| output activation suppression | $90.73 \pm 1.28$ | $0.15 \pm 0.38$ | $91.44 \pm 1.22$ | $1.05 \pm 1.13$ |
| both | $91.51 \pm 0.68$ | 0 | $93.96 \pm 0.60$ | $0.21 \pm 0.45$ |

## A.9 SEQUENTIAL MUTICLASS REMOVAL

This scenario demonstrates the practical use case of our algorithm where different unlearning requests come at different instances of time. In our experiments, we sequentially unlearn classes 0 to 8 in order from the CIFAR10 dataset on VGG11 and ResNet 18 model. The retain accuracy of the unlearnt model is plotted in Figure 9. The forget accuracy for all the classes in the unlearning steps was zero. We observe an increasing trend in the retain accuracy for both the VGG11 and ResNet18 models which is expected as the number of classes reduces or the classification task simplifies.

## A.10 MIA ATTACK DETAILS

The goal of the MIA experiment was to demonstrate how the unlearnt models behave as compared to the Retrained model and the original model. Below we mention the details of MIA experiments.

**Training** - We train a Support Vector Machine (SVM) classifier as a MIA model to distinguish between $D_{train\_r}$(as class 1 or member class) and $D_{test\_r}$(as class 0 or non member class).

**Testing** - We show this SVM model $D_{train\_f}$ to check if the MIA model classifies it as a member or non member. When the MIA model classifies it class 0 (Non Member) then the MIA model believes that the samples from $D_{train\_f}$ do not belong to the Train set. This is what is meant by having a high accuracy on $D_{train\_f}$.

**Interpretations of MIA scores**- We use the training and testing procedures mentioned above for all the models. Below we present interpretation for different models

- Original model - We see that the original model has a low MIA score (nearly 0) which means the SVM model classifies $D_{train\_f}$ as member samples. This is expected as $D_{train\_f}$ belonged to the training samples.

- Retrained model - We see that the Retrained model has a high MIA score ($100\%$) which means the SVM model classifies $D_{train\_f}$ as non members. This is expected as $D_{train\_f}$ does not belong to the training samples.

- Unlearnt model - By these experiments of MIA we wanted to see how MIA scores of unlearnt models perform. We observe the model unlearnt with our algorithm consistently performs close to the retrained model as compared to other baselines.

## A.11 VARIANTS OF OUR ALGORITHM

This section presents two variants of the algorithm depending on the location of the activation suppression. Consider the linear layer $a_o = a_i \times \theta^T$, where $a_o$ and $a_i$ are the input activation and output activations of a linear layer. The algorithm presented in the main paper focuses on activations before the linear layer, i.e. the input activations $a_i$. We could also suppress the output activations. This activation suppression meant projecting the parameters on the orthogonal discriminatory projection space $(I - P_{dis})$, which is post multiplying the parameters $\theta$ with $(I - P_{dis})^T$. Now if we were to suppress the output activations $a_o$ it would be the same as pre-multiplying the parameters $\theta$ with $(I - P_{dis})^T$. (Note, for suppressing $a_o$ the output activations are used to compute $P_{dis}$). This variant of our approach is capable of removing the information in the bias and normalization parameters of the network. The other variant suppresses both the input and output activations using their respective projection matrices. The results for these variants are presented in Table 7. We observe that the performance of these two variants is lower than the algorithm in the main paper and hence do not analyze it further.

