# OpenReview forum: "Deep Unlearning: Fast and Efficient Training-free Approach to Controlled Forgetting"
_ICLR.cc/2024/Conference — Submitted to ICLR 2024_

### Official Review · Reviewer_8Shn · 2023-10-28

**Soundness:** 3 good
**Presentation:** 2 fair
**Contribution:** 2 fair
**Rating:** 5
**Confidence:** 3

**Summary:**

This paper focuses on the question of whether we can unlearn one or more classes from a well-trained model if only a few samples are accessible from the training data of a large dataset. To address this question, the authors propose a novel singular value decomposition-based class unlearning method. This work first estimates the retain space and forget space of the model layer by layer based on the singular value decomposition technique, and then removes the shared information between spaces from the forget space to isolate the class-discriminatory feature space for unlearning, and finally, projects the model weights in the orthogonal direction of the class-discriminatory space to obtain the unlearned model. The authors demonstrate the effectiveness of the method on CIFAR-10, CIFAR-100, and ImageNet datasets as well as VGG11, ResNet18, and ViT models. They also display the applicability of their method in two practical scenarios of multi-class unlearning.

**Strengths:**

**Originality:** As far as I know, this paper is the first to propose the method of decomposing feature space through SVD to solve class unlearning, so this work is novel.

**Quality:** The approach seems reasonable. By decomposing the feature space of each layer of the well-trained model, the features of the unlearn class are eliminated while reducing the impact on the features of the retained classes. The experimental evaluations in the paper also provide evidence to support the claims made in the paper. In particular, there are verifications on the large datasets ImageNet and ViT. However, the method also has some unclear aspects that I mentioned below.

**Clarity:** The paper is generally well-written. However, there is room for improvement as I mention below.

**Weaknesses:**

**Methodology:**
1. The authors only describe the representations collection of linear and convolutional layers, but how to deal with the transformer layer, BatchNorm, etc.?
2. The authors perform SVD operation based on the representations matrix of $X_r$ and $X_f$. Do the sample size and sampling strategy of $X_r$ and $X_f$ affect the results of the algorithm?
3. Why did the authors consider designing importance-base space scaling? Won't this cause the deformation of the forget space and retain space?
4. The choice of best $\alpha$ in Eq.(1) seems to be a trick as different datasets give different $\alpha$ sets. Is there a more reasonable way to choose, like learnable $\alpha$?


**Experiments:**
1. The authors mention that the experimental results come from 10 different target unlearning classes, and CIFAR-100 is evaluated for every 10th. Does it mean that the [10, 20, …, 100]-th class was used as an unlearning class to conduct the experiment?
2. Does the "Original" represent the well-train model? Are the results reported on all retain and forget classes?
3. How is the classification head designed? For example, for CIFAR-100, the classification head of the well-train model outputs a 100-dimensional vector. What about the unlearning model?
4. In Figure 4, the proposed method does not have an advantage in efficiency compared to the Tarun et al. (2023)? Could the authors provide further analysis?
5. While I think the multi-class forgetting experiment is interesting, what the authors provide is not sufficient. First, simply conducting experiments on CIFAR10 is not convincing. Second, the effects of baselines were not compared. Third, the result analysis is insufficient. For example, why is the original model of resnet18 better in Figure 5a, but not as good as the unlearning model in Figure 5b?

**Questions:**

1. See weakness for details on methodological and experimental issues.
2. The authors only mention three related works of literature on class unlearning. I am wondering about the importance and practicality of this problem.
3. This paper only evaluates the effectiveness of the method based on the classification accuracy of forgetting and retaining classes. Have the authors considered verification metrics with more theoretical guarantees? If the forgetting class is known, the output probability of the forgetting class can be forced to 0 without changing the model parameters to ensure the effect of forgetting and retaining classes, but this approach is obviously contrary to the motivation of unlearning.

---

> ### Author Response · Authors · 2023-11-19
> **Reply to Reviewer 8Shn**
>
> **Response to Weakness:**
> - **Methodology**
>     1. We have added a line in section 4.3 under “Discussion” to clarify this.  We apply our algorithm to all the linear layers in the transformer architecture. For an attention block, the linear layer weights updated are the Key Weights, Query Weights, Value Weights, and Output weights. We do not perform any unlearning of the normalization layers as the fraction of total parameters for these layers is insignificant and empirically is not seen to affect the performance of our algorithm.
>     2. Samples from each class are necessary to ensure a good estimation of retain space and forget space. We chose $X\_r$ and $X\_f$ to be around 1000 to ensure we have at least 1 sample from each class (ImageNet). Empirically we observed that having 1000 samples was enough and we did not increase it to ensure the compute cost of Representation collection and SVD over $X\_r$/$X\_f$ is kept low.
>     3. In our algorithm  (line 14, Algorithm 1) we find $P\_{dis}$, which is the class discriminatory activation space. We want to find a class discriminatory space where forget samples have the ‘highest’ activations whereas retain samples have the 'lowest'. By removing this space (from the entire space) we perform our unlearning. To assign this connotation of ''highest/lowest' activation with the original $P\_f$/$P\_r$ space we need to assign importance - thus scaling was used.  We agree this causes deformation of the space - which is a design choice.
>     4. Our algorithm is gradient free algorithm and obtains the parameter update in a one-shot fashion. As we did not use gradient information it was not straightforward to come up with a learning scheme for $\alpha$. However, in the future training algorithms can be designed that could learn $\alpha$ to be used for unlearning if needed.
> - **Experiments**
>     1. For CIFAR100 we use [1, 11,21, … 91 ]. We update the paper stating  “CIFAR10 dataset is accessed for unlearning on each class and CIFAR100 is evaluated for every 10th starting from the first class.”
>     2. Yes, original refers to a well trained model which is trained on $D\_{train}$. The accuracy numbers in the previous version were the overall accuracy of the network on the samples of retain and forget class. We have updated these numbers to report retain class accuracy and forget class accuracy separately.
>     3. To evaluate the unlearnt model we assume that the unlearning algorithm gives us the unlearnt model without changing the classifier output neurons similar to the baseline SoTA methods Tarun et al and Kurmanji et al.
>     4. We thank the reviewer for pointing this out. We realized that the runtimes depend on the implementation of the algorithm and the background hardware it is running on. We analytically find the compute costs of different algorithms and plot them in Figure 5 and show the details in Appendix A.7. Our algorithm has at least 6.5x lower compute than other methods.
>     5. We present the results for superclass unlearning on CIFAR100 in Figure 6 and have moved the results of CIFAR10 to Appendix A.8. Further, we have added a comparison with baselines in Table 6 in Appendix A8. We see our model performs better on MIA scores and retain accuracy over SoTA Tarun et al. NegGrad+ is seen to outperform our method on ResNet18. We saw Kurmanji et al to have extremely low retain accuracies and hence have not reported them in the table. There was a mistake in the plot of Figure 5b in the original submission (10b in updated). We did not plot the original model’s forget accuracy in the figure.
>
> **Response to Questions:**
>
> 2. Class unlearning is important where entire user data belongs to a single class. Consider a scenario where we train a classification model for face recognition. In this case, an entire class belongs to a single person and we would be required to remove this class when on request. Further, in the cases where a collaborative multi-task model is shared by a few companies and one of the companies decides to withdraw its data the one-shot multi-class removal algorithm becomes relevant.
> 3. We have added membership inference attack evaluations in our experiments. The results show our method performs well across different datasets and architectures obtaining an average improvement of 7.8% in MIA accuracy over other baselines.
> Our MIA attack has access to the true confidence scores of the model on the correct class. Hence an algorithm that post hoc manipulates the output would fail to obtain good MIA scores.
>
> **Note:** We have added a version of the paper with highlighted changes to supplementary material.

---

> ### Author Response · Authors · 2023-11-22
> **Looking forward to feedback**
>
> Dear Reviewer 8Shn,
>
> We appreciate your valuable comments on our work. We have updated the paper as per your suggestion addressing the weaknesses and questions. We look forward to any additional feedback you have for improving the quality of our work further.
>
> Thanks
> Authors

---

### Official Review · Reviewer_FQb4 · 2023-10-29

**Soundness:** 2 fair
**Presentation:** 3 good
**Contribution:** 2 fair
**Rating:** 5
**Confidence:** 4

**Summary:**

This paper proposes a novel method for class unlearning inspired by work in continual learning that uses Singular Value Decomposition as a means for separating ‘spaces’ containing knowledge of different tasks. In this work, aninstantiation of SVD is used to separate out the ‘forget space’ (the ‘space’ in which the activations of samples belonging to the forget set lie) from the ‘retain space’ (analogously, for retain samples). These spaces are obtained by running SVD on activations (retain or forget) from all layers of the network. The authors show that a small subset of the retain and forget sets suffices for obtaining these activations. Once the forget space is identified, the model weights are updated by a projection to a space that removes forget set information.
The authors also use a baseline that they refer to as Stable Ascent (which was proposed in recent work). Interestingly, they show that both this baseline and their proposed method surpass the previous state-of-the-art in the context of class unlearning on some benchmarks, with their proposed method making further progress over Stable Ascent. They investigate empirically different scenarios (different datasets and architectures, removal of one or more classes either in one-go or sequentially) and report both quantitative results (accuracies) and qualitative ones (gradcam heatmaps to visualize feature saliency).

**Strengths:**

- The paper studies the important problem of unlearning that is attracting increasing attention recently
- The proposed method is well-motivated and an interesting idea
- For the most part, the paper is well written (see below for some exceptions)
- Indeed the Transformer results are the first to my knowledge application of unlearning methods in larger-scale models that are closer to the state-of-the-art, which is really interesting.
- interesting qualitative analysis of saliency.

**Weaknesses:**

- Motivation: the way that the problem of unlearning is motivated in this paper (data deletion; user privacy) seems at odds with the problem of unlearning classes (as it would correspond to unlearning individual data points that don’t necessarily belong to the same ‘class’, depending on the definition of class). What are application scenarios for class unlearning? While the authors have motivated the problem of unlearning well, motivation of missing for class unlearning in particular.
- the authors claim that Stable Ascent is one of the contributions of the paper but this baseline has already been proposed in previous work that the authors did not cite ([A] – see References below, where it is referred to as NegGrad+ and the authors of that work also find that it is a strong baseline, surpassing previous SOTA in several scenarios)
- recent unlearning methods are missing from the Related Work section, e.g. [A, B, C, D] (see References below), and it would also significantly strengthen the paper to empirically compare against them too.
- ablations are missing. For example, how large is the contribution of the proposed scaling? It would be good to investigate a version of the proposed method without this. Further, how important is it to use activations from all layers for SVD versus just the top layer(s)?
- also, it would be good to motivate the scaling a bit more. Is such a scaling used in the continual learning literature / related methods? If not, what is different about this application that necessitates it?
- the evaluation is lacking. While several evaluation metrics are used for unlearning, the authors rely primarily on accuracy metrics. A particularly important class of evaluation techniques that is missing is membership inference attacks (see e.g. the papers by Golatkar et al, which are cited in this work, and also see [A] from the references below)
- In fact, the evaluation metrics seem to be at odds with the goal of class unlearning that the authors state in the Introduction, namely that “the unlearning algorithm should produce parameters that are equivalent to those of a model trained without the target class”. Despite this definition, the authors don’t look at proximity in weight space or related metrics and instead rely primarily on accuracy.
- In section 3, the description of the problem of class unlearning isn’t precise enough. It’s defined as producing a set of unlearned parameters such that two conditions (test retain examples are correctly classified and test forget examples incorrectly classified) are satisfied for ‘many samples’. But how many samples? Usually, the accuracy on each of those two sets is desired to be just as high/low as it would be for retrain-from-scratch. Is there a reason that this is not the definition used? Also, how come this definition refers only to test examples? Usually it is also desired to have similar conditions hold for the training set (retain and forget partitions).
- clarity: the algorithm is presented in terms of linear and convolutional layers. But in their experiments, the authors also use Transformers. It’s not directly obvious how the proposed method is used for attention layers.
- clarity: “for both linear and convolutional layers, where l is a layer and i is retain sample ” – i was not mentioned in that context.

Minor issues and typos
==================
- ‘produce parameters that are equivalent to those of a model trained without the forget set’ – not clear what the word ‘equivalent’ means here.
- ‘Generalization on retain samples’ – this is a little confusing as the retain set is part of the training set, so generalization isn’t an appropriate term (as it refers to held-out samples). ‘accuracy’ or ‘performance’ are more appropriate.
- ‘we asks the question’ → ‘we ask the question’
- ‘this section focus’ →‘this section focuses’
- ‘sorted in by the amount →‘sorted by the amount’
- ‘the samples form class to be retained’ →‘the samples from the class’
- In several places, some articles are missing, and there are more typos than these mentioned here.
Please proofread the paper and check for grammatical errors.

References
==========
- [A] Towards Unbounded Machine Unlearning. Kurmanji et al. NeurIPS 2023.
- [B] Towards Adversarial Evaluations for Inexact Machine Unlearning. Goel et al. 2023.
- [C] Prompt Certified Machine Unlearning with Randomized Gradient Smoothing and Quantization. Zhang et al. NeurIPS 2022.
- [D] Unrolling SGD: Understanding Factors Influencing Machine Unlearning. Thudi et al. 2022

**Questions:**

- What is it about this method that makes it specific to class unlearning? Could one use this method to unlearn a random subset of the training dataset? If not then why not?
- in figure 4, I was surprised that the method of Tarun et al. is more efficient than the proposed method, since Tarun et al uses SGD (in several phases too) while the proposed method only requires forward passes of a few samples (and no backward passes). Why is that?
- in figure 5b, why are the confidence intervals so large? Can we draw any meaningful conclusions from this figure?
- why is the reference point / oracle of Retraining not included in all tables? Is this because it is too computationally expensive to compute this for some models / datasets? Without that reference point, it is not possible to know what the target accuracy/error is for the forget set (because the goal is usually defined as matching / being as close as possible to the accuracy/error of Retrain on the forget set). Could you explain how the results are interpreted given the absence of that reference point?

---

> ### Author Response · Authors · 2023-11-19
> **Reply to Reviewer FQb4**
>
> **Response to Weakness:**
> 1. **Motivation for Class Unlearning -** Class unlearning is important where entire user data belongs to a single class. Consider a scenario where we train a classification model for face recognition[1]. In this case, an entire class belongs to a single person and we would be required to remove this class on request. Further, in the cases where a collaborative multi-task model is shared by a few companies where a task is classification over a group of classes and one of the companies decides to withdraw its data, the multi-class removal algorithm becomes relevant.
> 2. **NegGrad+ baseline-** We thank the reviewer for pointing this out. We have added the citation and updated the contributions. Further, we have renamed Naive Ascent as NegGrad and Stable Ascent as NegGrad+ to be consistent with the literature.
> 3. **Comparisons-**  We add experiments on [kurmanji et al] (or reference [A] in the Reviewers list) as it is a more recent method in the list of suggestions. We have updated the “Comparisons” part in Section 5 mentioning this as a baseline and added the results in Table1 and Table2. We see our method has 13.6% better forget accuracy and 45.7% better MIA over [A] on average across all the datasets and networks in Table 1 and Table 2.
> 4. **ablations are missing-**
>     -  **Proposed scaling:** Without the proposed scaling the $\Lambda\_r$ and $\Lambda\_f$ in line 12 of Algorithm 1 become Identity. Now as $U$ matrix obtained through SVD is an orthonormal matrix, $P\_r$ and $P\_f$ in line 13 of Algorithm 1 become Identity when $\Lambda\_r$ and $\Lambda\_f$ are Identity. When $P\_r$ and $P\_f$ is Identity $P\_{dis} = P_f(I-P\_r)$ computed in line 15 becomes a zero matrix. Projection of weights on $I-P\_{dis}$ in line 15 would hence cause no change in the weight as $I-P\_{dis}$ is Identity essentially leading to no unlearning. This necessitates the scaling in line 12 of the algorithm. We have clarified this under the “Importance-base Space Scaling” in Section 4.2.
>     -  **Impact of layer:** We have added a section on “Layer-wise analysis” where our results show that later layers play a more significant role in unlearning. Further, in Figure 8 of Appendix A.6 we see that removing our algorithm from the first 4 layers has a very low impact on forget accuracy.
> 5. **Scaling Motivation-** In continual learning, scaling has been used to modulate the gradient descent steps during training so that old tasks are not forgotten and new tasks are learned with high accuracy.  In this scenario, we apply one-shot scaling of the weights on a pre-trained model. Effectively, we apply scaling to restrict the layer-wise activation of the input samples (during inference) to remove activations belonging to class discriminatory space $P\_{dis}$. Further, we have added a few lines to explain the need for importance scaling under “Importance-base Space Scaling” in Section 4.2.
> 6. **MIA attacks-** As per reviewer’s suggestion, we have added membership inference attack evaluations in our experiments. The results show our method performs well across different datasets and architectures obtaining an average improvement of 7.8% in MIA accuracy over other baselines.
> 7. **Class Unlearning goal in Introduction-** We meant “The unlearning algorithm should produce model (parameters) that are functionally equivalent to those of a model trained without the target class.” by the statement and have updated this in the paper. We show such equivalence by our experiment using ACC_r, ACC_f, MIA, confusion matrix, and gradient based saliency maps.
> 8. **Problem description in Section 3-**  Thanks for this suggestion. We have modified the writing to address reviewers concern.
> We clarify the definition in the following places in the updated paper.
> Class Unlearning  ( Section 1 Introduction) “For a class unlearning setup, the primary goal of the unlearning algorithm is to eliminate information associated with a target class from a pretrained model. This target class is referred to as the forget class, while the other classes are called the retain classes.” Additionally, we have formally presented “class unlearning” in Section 3: “The parameters $\theta^*\_f$ must be functionally indistinguishable from a network with parameters $\theta^*$, which is retrained from scratch on the samples of $\mathcal{D}\_{train\_r}$ in the output space. In other words, these parameters must satisfy $f(x\_i, \theta^*) \simeq f(x\_i, \theta^*\_f)$ for $(x\_i, y\_i) \in D\_{test}$ or $D\_{train}$.”
>
> **Note:** We have added a version of the paper with highlighted changes to supplementary material.
>
> Reference:
> [1] Y. Sun, X. Wang, and X. Tang, "Deep learning face representation from predicting 10,000 classes." pp. 1891-1898.

---

> ### Author Response · Authors · 2023-11-19
> **Reply to Reviewer FQb4**
>
> **Response to Weakness(Continued):**
>
> 9. **Transfomer Layers -** We have added a line in section 4.3 under “Discussion” to clarify this.  We apply our algorithm to all the linear layers in the transformer architecture. For an attention block, the linear layer weights are the Key Weights, Query Weights, Value Weights, and Output weights. We do not perform any unlearning of the normalization layers as the fraction of total parameters for these layers is insignificant and empirically is not seen to affect the performance of our algorithm.
> 10. **typo-** We updated the line to  “ for both linear and convolutional layers, where $l$ is layer.”
>
> **Response to Questions:**
>
> 1. Our method relies on SVD which analyzes the underlying statistics of the activation spaces. We believe our method would work on any unlearning problem where there are differences in the data distributions of the forget samples and the retain samples. Class Unlearning is one such case where there is a significant difference in the data distributions, which makes it easier for SVD to figure out distinct spaces. When we assign a random subset of training samples as forget set, SVD can not distinguish well between the activations of forget samples and retain samples. This is because Retain samples are also drawn similarly (Independently Identically Distributed) from the training set. This is why the SVD based method would need further modifications to be fully effective for random subset unlearning problems.
> 2. We thank the reviewer for pointing this out. We realized that the runtimes depend on the implementation of the algorithm and the background hardware it is running on. We analytically find the compute costs of different algorithms and plot them in Figure 5 and show the details in Appendix A.7. We see 7.3x lesser compute for our approach than the baseline.
> 3. The mean forget accuracy in Figure 5b in the original version (it has been moved to Figure 10 in the updated version) is less than 0.37 on average. In all the experiments we ran for 5 class unlearning the forget accuracy was 0 in 6 of the 10 experiments (the other experiments have forget accuracy of 0.32, 0.7, 0.74, 1.88). In the experiments for 2 class unlearning 9 of the 10 experiments have 0 forget accuracy while one experiment had a forget accuracy of 2.5. These experiments show very low forget accuracies on the model and hence we believe are significant.
> 4. Yes, it was computationally expensive to compute these numbers for ImageNet dataset. To address reviewer concern we have updated Section 6 with lines “Due to the training complexity of the experiments, we were not able to obtain retrained models for ImageNet. We observe that the results on CIFAR10 and CIFAR100 datasets consistently show $ACC_f$ to be 0 and the $MIA$ performance to be $100\%$. We, therefore, interpret the model with high $ACC_r$, $MIA$, and low $ACC_f$ as a better unlearnt model for these experiments.”
>
> **Response to Minor Issues and typos**
> - We thank the reviewer for pointing this out. We went over the document and corrected grammatical errors.

---

> > ### Comment · Reviewer_FQb4 · 2023-11-21
> > **thank you for the responses**
> >
> > Hi authors, thank you for the detailed responses to my comments!
> >
> > I think the new version is improved due to the clarifications (e.g. the definition of the problem, the importance of the scaling, how the proposed methods is used in the transformer architecture, runtime computation, etc), additional analyses (e.g. layer-wise analysis), experiments (the MIA results in particular).
> >
> > I have a question about the MIA results. The authors state "We report the average model predictions on Dtrain_f as the MIA scores in
> > our evaluations. A high value of MIA score for a given sample indicates that it does not belong to the training data.". I'm not sure I understand the rationale behind this way of computing a 'score'. Notably, having the MIA attacker be *certain* that the forget data was not seen is not necessarily the intended behaviour? Intuitively, we may want it to only be as certain about this as the attacker would have been if it was attacking a retrain-from-scratch model.
> >
> > In prior works, the goal for MIA evaluations has been phrased as either having the binary MIA attacker have an accuracy of 50% (which, assuming that the set of members and non-members it is queried on are balanced, indicates that the attacker cannot distinguish the forget set from non-members). Could you comment on this difference? Generally, it would be great to precisely describe the MIA setup (on what data exactly was the SVM trained on and what scores we expect the ideal unlearner to have).
> >
> > I also think the authors should explicitly discuss the limitations of their method in the paper (as they say, by design this approach is perhaps more appropriate for class unlearning / situations with pronounced differences in the distributions of the forget and retain sets).

---

> > > ### Author Response · Authors · 2023-11-22
> > > **Response to additional questions by Reviewer FQb4**
> > >
> > > 1. We thank the reviewer for pointing this out. We have added a line under Evaluation in Section 6 - “An unlearnt model is expected to match the $MIA$ score of the Retrained model (See Appendix A.10).” The goal of this experiment was to demonstrate how the unlearnt models behave as compared to the Retrained model and the original model. Below we mention the details of MIA experiments and how MIA score is defined/computed.
> > >
> > >     **MIA model evaluation procedure-**
> > >
> > >     **Training-** We train a Support Vector Machine (SVM) classifier as an MIA model to distinguish between $D\_{train\_r}$(as class 1 or member class) and $D\_{test\_r}$(as class 0 or nonmember class).
> > >
> > >     **Testing-** We show this SVM model $D\_{train\_f}$ to check if the MIA model classifies it as a member or nonmember. We report this classification accuracy as MIA score in the paper. When the MIA model classifies it as class 0 (Nonmember), then the MIA model believes that the samples from $D\_{train\_f}$ do not belong to the Train set.
> > >
> > >     **Experiments/Interpretations of MIA scores**- We use the training and testing procedures mentioned above for all the models. Below we present interpretations for different models
> > >     - Original model -  We see that the original model has a low MIA score (nearly 0) which means the SVM model classifies $D\_{train\_f}$ as member samples. This is expected as $D\_{train\_f}$ belonged to the training samples.
> > >     - Retrained model - We see that the Retrained model has a high MIA score (100%) which means the SVM model classifies $D\_{train\_f}$ as nonmembers. This is expected as $D\_{train\_f}$ does not belong to the training samples.
> > >     - Unlearnt model - By these experiments of MIA we wanted to see how MIA scores of unlearnt models perform. In particular, is it close to the behavior of Original model or Retrained model?
> > >
> > >     We train the MIA model to distinguish the train and test samples. We believe studying how the $D\_{train\_f}$ samples are classified by the MIA model reveals significant information about the behavior unlearnt model. Further, as we evaluate all the models in a similar fashion we can compare them against each other.
> > >
> > > 2. We have added a line Under “Discussion” stating the limitation “Further, our algorithm assumes a significant difference in the distribution of forget and retained samples for SVD to find distinguishable spaces. This is true in the class unlearning setup, where retain and forget samples come from different non-overlapping classes, thus our methods archive SoTA results here. However, in the case of unlearning a random subset of training data, this assumption would not hold. Thus our method has limited performance in such a scenario and requires additional modification for effective unlearning.”
> > >
> > > We thank the reviewer for the timely response to our rebuttal and are grateful for the reviewer's feedback on improving the quality of our work. We would be happy to answer any further questions the reviewer might have.

---

> > > > ### Comment · Reviewer_FQb4 · 2023-11-22
> > > > **thank you for the clarifications, and some follow-up thoughts**
> > > >
> > > > Thank you, the MIA procedure used is now clearer. It would be great to also update the paper with these details.
> > > >
> > > > Some follow-up thoughts on this: if the forget set has a distribution that is quite distinct from the retain set distribution (as is the case for class unlearning for instance), then it is perhaps easier for a "mia" classifier to not think that the forget set belongs to the training set (merely because it follows a different distribution!). So, in this particular case, what the "mia" classifier may be doing is detecting the distribution change rather than actually performing (solely) membership inference.
> > > > it would be great to explicitly state in this context which classes are part of the retain and test sets that are used to train the binary mia classifier; and which class(es) are part of the forget set that this binary classifier is then queried on, to surface this issue.
> > > >
> > > > With this in mind, perhaps applying LiRA-like MIAs ("membership inference attacks from first principles", Carlini et al) is more appropriate, which operate on a per-example basis, and model the distribution of losses/confidences of *that particular example* when it is included vs excluded from training.
> > > >
> > > > However, I do think the experiments that the authors conducted are a great first step in this direction and already strengthen the paper.

---

> > > > > ### Author Response · Authors · 2023-11-22
> > > > > **Response to follow-up by reviewer FQb4**
> > > > >
> > > > > We have added the MIA details to Appendix A10 of the paper and referred to it in Section 5 under “Evaluation”. Further, we have added the marked changes for the 2nd update in the supplementary.
> > > > >
> > > > > **Clarification on retain and Forget set for MIA -**
> > > > > 1.  For each experiment we individually train the MIA model. The retain and the forget set depend on the experiment under consideration.
> > > > > 2. For example, our experiments in Table 1 with VGG11_BN and CIFAR10 have 10 experiments (unlearning each class 0, 1, 2 …9  ). Consider the case of unlearning class 0 (airplane class), the MIA model would have classes 1-9 in retain set and class 0 in forget Set. The MIA score for this particular experiment is calculated using the procedure we described in our previous comment (details in Appendix A10). Note that the notations used in Appendix A10 are consistent with notation of Section 3. In particular the training data for the MIA model would be the train samples of class 1-9 as member samples and test samples of class 1-9 as non member samples. The forget Set or training samples of class 0 is used for inference.
> > > > > 3. This is repeated for all the 10 experiments to get an average number for the MIA score in Table 1 for VGG11 on CIFAR10 dataset. Similar experiments are performed for all the dataset and models to get MIA scores for Table 1 and Table2.
> > > > >
> > > > > **LiRA-like MIAs -**
> > > > >
> > > > > 1.  We agree with reviewers' suggestion that a LiRA-like MIA attack would further strengthen the paper. However these MIA attacks require training multiple models on different subset of training data to obtain the distributions of losses/confidence for each sample when it is included vs excluded for training. This would be extremely compute intensive in the case of datasets like ImageNet used in our experiments. Hence we used simple MIA attacks in our work.

---

### Official Review · Reviewer_bRqF · 2023-10-31

**Soundness:** 3 good
**Presentation:** 3 good
**Contribution:** 2 fair
**Rating:** 3
**Confidence:** 5

**Summary:**

The paper proposes a method for unlearning an entire class or a group of classes from a learned model. They estimate internal activations corresponding to all the layer, for the forget and retain set and then compute a forget and retain space. Then they intersect the two spaces and removes it from the forget space, and final project the weights onto the orthogonal space corresponding to this space. They show that such a method is efficient and performs better than the contemporary methods.

**Strengths:**

1. The paper address the problem of machine unlearning which is an important problem given the recent explosion in large scale models.
2. The proposed method is easy to use, as it applied layer wise, and only requires layer wise SVD computation.
3. Stable Ascent as a heurestic based method for unlearning is interesting in the case of linear models, however, in this case it is for non-linear models.
4. The paper provides empirical results for different datasets and models.

**Weaknesses:**

1. The paper is lacking a clear and precise definition of unlearning. Its is important to show the definition of unlearning that you want to achieve through your algorithm.
2. The proposed algorithm is an empirical algorithm without any theoretical guarantees. It is important for unlearning papers to provide unlearning guarantees against an adversary.
3. The approach is very similar to this method (http://proceedings.mlr.press/v130/izzo21a/izzo21a.pdf) applied on each layer, which is not cited.
4. A simple baseline is just applying all the unlearning algorithm mentioned in the paper to the last layer vs the entire model. This comparison is missing.
5. All the unlearning verification are only show wrt accuracy of the model or the confusion matrix, however, the information is usually contained in the weights of the model, hence other metrics like membership attack or re-train time after forgetting show be considered.
6. The authors should also consider applying this method a linear perturbation of the network, as in those settings you will be able to get theoretical guarantees in regards to the proposed method, and also get better results.
7. Since the method is applied on each layer, the authors should provide a plot of how different different weights of the model move, for instance plot the relative weight change after unlearning to see which layers are affected the most after unlearning.

**Questions:**

Please see the weaknesses.

---

> ### Author Response · Authors · 2023-11-19
> **Reply to Reviewer bRqF**
>
> **Response to Weakness:**
> 1. Thanks for this suggestion. We have modified the writing to address this.
> We clarify the definition in the following places in the updated paper.
> Class Unlearning  ( Section 1 Introduction) “For a class unlearning setup, the primary goal of the unlearning algorithm is to eliminate information associated with a target class from a pretrained model. This target class is referred to as the forget class, while the other classes are called the retain classes.” Additionally, we have formally presented “class unlearning” in Section 3: “The parameters $\theta^*\_f$ must be functionally indistinguishable from a network with parameters $\theta^*$, which is retrained from scratch on the samples of $\mathcal{D}\_{train\_r}$ in the output space. In other words, these parameters must satisfy $f(x\_i, \theta^*) \simeq f(x\_i, \theta^*\_f)$ for $(x\_i, y\_i) \in D\_{test}$ or $D\_{train}$.”
> 2. We pose our work as empirical and evaluate our work on large-scale experiments with various sized dataset (including ImageNet) and SoTA deep neural network models (e.g. ViTs etc’). The SoTA baselines we compare against [Tarun et al] and [kurmanji et al] are also empirically evaluated and the scale of their experiments is limited. Further, We have added experiments for Membership Inference Attacks to demonstrate the efficacy of forgetting.
> 3. We thank the reviewer for providing the reference. We have added reference to this work in our paper. We would like to point out the fundamental differences between this work and our work.
>     -  It uses synthetic data and gradient-based optimization/finetuning of the model on these data for unlearning. Whereas we do not use any synthetic data and we don’t need any gradient-based finetuning of the given pre-trained model.
>     - The scale of experiments is very different. In particular, this work proposes data deletion with a particular focus on linear and logistic regression whereas we propose an unlearning method for deep learning models that give SoTA performance.
>     - At its core this work relies on being able to calculate the logits of the retrained model without having access to the weights. Essentially calculating  $f(x\_i,\theta^*)$ without knowing $\theta^*$, where $x\_i$ is a forgot sample and $\theta^*$ is retrained model weights. This is a challenging problem to solve in the case of deep learning models and datasets where data might not be very separable which limits the algorithm's application in deep networks. Our algorithm does not rely on the knowledge of logits of the retrained model and hence easily scales for challenging datasets and networks.
> 4. We add the results for applying our method to the top layer in Appendix A6. Figure 8 shows applying our approach to the last layer only (indicated by layer=10) has a forget accuracy 65% which indicates that removing information from other layers is also necessary.
> 5. We have added membership inference attack evaluations in our experiments. The results show our method performs well across different datasets and architectures obtaining an average improvement of 7.8% in MIA accuracy over other baselines.
> 6.  Can the reviewer clarify this point e.g. what is linear perturbation of the network in the context of unlearning? Some reference would be helpful.
> 7. We have added this plot in Figure 4. We see that later layers are more affected by unlearning through our approach. This is expected as the later layers are expected to learn complex class discriminatory information while the initial layers learn edges and simple textures [1].
>
> **Note:** We have added a version of the paper with highlighted changes to supplementary material.
>
> Reference:
>
> [1] Chris Olah, Alexander Mordvintsev, and Ludwig Schubert. Feature visualization. Distill, 2017

---

> ### Author Response · Authors · 2023-11-22
> **Looking forward to feedback.**
>
> Dear Reviewer bRqF,
>
> We appreciate your valuable comments on our work. We have updated the paper as per your suggestion addressing most of the weaknesses and questions. We look forward to any additional feedback you have for improving the quality of our work further.
>
> Thanks
> Authors

---

### Official Review · Reviewer_ZSgK · 2023-11-01

**Soundness:** 3 good
**Presentation:** 3 good
**Contribution:** 3 good
**Rating:** 8
**Confidence:** 4

**Summary:**

In this research endeavor, a novel class unlearning algorithm is introduced, meticulously designed to eliminate an entire class or a group of classes from the acquired model. The algorithm, developed within this study, initiates by estimating two essential spaces: the Retain Space and the Forget Space. These spaces represent the feature or activation spaces corresponding to the samples from classes that need to be retained and unlearned, respectively. The method proposed for obtaining these spaces leverages a unique singular value decomposition-based technique, mandating the collection of network activations at different layers via several forward passes through the network. Subsequently, the shared information between these spaces is computed and selectively removed from the Forget Space, thus isolating the class-discriminatory feature space for the unlearning process. Ultimately, the model weights are projected in the orthogonal direction of the class-discriminatory space, resulting in the derivation of the unlearned model.

**Strengths:**

1. The method is very simple and elegant.
2. Provides a strong baseline stable-ascent and also the proposed method beats the stable ascent.
3. Results are more satisfactory than current SoTA methods.

**Weaknesses:**

1. Requires a few training samples for unlearning. There are methods for zero-shot unlearning.

**Questions:**

1. There is a more fundamental question about the class unlearning setup. The whole point of unlearning is to replicate a completely retrained model in parameter space or in output space  Now for the preliminary section 3 it is mentioned that for an unlearned model the output label of a datapoint belonging to an unlearned class, is not a true label in this case the unlearned class i.e. y_i != f(x_i,\theta_f). Why is this the case? Is it not that the unlearned model output should be exact/almost the same as the retrained model? If the retrained model gives output as the true label on very few samples why unlearned model can’t give output the same? In other terms,. if the retrained model gives accuracy let's say 3% on the test forget set and the unlearning model also gives similar accuracy on the forget set. The unlearning method is valid. So, I think the y_i != f(x_i,\theta_f) can be formulated in better probabilistic terms so that it matches the retrained model. The implicit assumption is that for a retrained model the accuracy on the forget set if 0 is not correct. We can set adversarial examples such that the retrained model gives an accuracy of 100% on the forget set.

---

> ### Author Response · Authors · 2023-11-19
> **Reply to Reviewer ZSgK**
>
> **Response to Weakness**:
> 1. The reviewer is right to point out there are zero-shot unlearning works. However, here zero-shot does not mean that they don't use any samples for unlearning. It means the algorithm first generates a few synthetic samples and then unlearns with those [1], usually with SGD based training/finetuning steps. On the other hand, we propose a novel SVD-based approach where we use only a few samples from the training set to get the forget space and modify the weights, without any training, to forget the desired class. Generating synthetic samples from the trained model is a well-established idea. We believe our SVD-based space estimation can be used on these synthetic data to get the desired forget space. Hence would enable zero-shot unlearning. We leave this exploration for the future.
>
> **Response to Questions**:
> 1. We have rewritten the definition in Section 3(Class Unlearning) as "The parameters $\theta^*\_f$ must be functionally indistinguishable from a network with parameters $\theta^*$, which is retrained from scratch on the samples of $\mathcal{D}\_{train\_r}$ in the output space. In other words, these parameters must satisfy $f(x\_i, \theta^*) \simeq f(x\_i, \theta^*\_f)$ for $(x\_i, y\_i) \in D\_{test}$ or $D\_{train}$." We thank the reviewer for pointing this out.
>
> **Note:** We have added a version of the paper with highlighted changes to supplementary material.
>
> Reference:
>
> [1] Chundawat, Vikram S., et al. "Zero-shot machine unlearning." IEEE Transactions on Information Forensics and Security (2023)

---

### Author Response · Authors · 2023-11-20
**Summary of Changes.**

We thank the reviewer for constructive feedback. Below is the list of changes we made to our work.
1. We clarified the definition of class unlearning In the Introduction - “The unlearning algorithm should produce model (parameters) that are functionally equivalent to those of a model trained without the target class.”
2. We corrected the formal definition of class unlearning in Section 3 - “The parameters $\theta^*\_f$ must be functionally indistinguishable from a network with parameters $\theta^*$, which is retrained from scratch on the samples of $\mathcal{D}\_{train\_r}$ in the output space. In other words, these parameters must satisfy $f(x\_i, \theta^*) \simeq f(x\_i, \theta^*\_f)$ for $(x\_i, y\_i) \in D\_{test}$ or $D\_{train}$.”
3. We have added a few lines in Section 4.2 to motivate scaling.
4. We have added a discussion on the transformer and normalization layers in section 4.3.
5. We renamed Naive Ascent to NegGrad and Stable Ascent to NegGrad+ to be consistent with the literature.
6. We have added Membership Inference attack to evaluate our models.
7. We added a recent work by Kurmanji et al to our experiments.
8. We have included results showing the layer-wise distance of weights(Figure 4), the effect of applying our algorithm to the top few layers (Figure 8), Compute analysis (Figure 5), and SuperClass removal on CIFAR100 (Figure 6 ).

---

> ### Author Response · Authors · 2023-11-22
> **Additional Changes**
>
> Below are the changes made in the second update of the paper.
>
> 9. We have added the limitation of our method in Section 4 under “Discussion”
> 10. We have added details on MIA in Appendix A10 and clarified interpretation of MIA scores in Section 5 under “Evaluation”
>
> **Note :** We have added a highlighted version of both the updates in supplementary  material.
>
> We thank all the reviewers for their feedback. We have made changes to our work as per their suggestions to improve our work.

---

### Meta-Review · Area_Chair_7WgH · 2023-12-07

**Metareview:**

The paper presents a novel way to unlearn classes from given pre-trained deep neural network models. Specifically, the proposed techniques makes use of "retain" and "forget" example sets, which are then used to compute singular value decomposition (SVD) on the respective activations, followed by orthogonal projection to "remove" the parts to be unlearned across all individual layers. Experimental evidence shows the effectiveness of this method over baselines that re-train models or fine-tune them through gradient-based methodology on CIFAR and ImageNet datasets for convolutional and (vision) transformer architectures.

All reviewers have commended the novelty of the work and are in consensus on the strengths of the paper. In particular, they appreciate the simplicity of the method given the strong empirical performance. However, they also raised several initial concerns, in a broad range of points from clarifications, definitions, motivation, missing explanations, request for additional empirical quantification and discussion.
During the discussion phase, the authors have made several modifications and extensions to the appendix of the paper in an attempt to clarify these points. Here, notable improvements include the addition of memberships inference attacks as an additional measure to quantify the unlearned parts (rather than just reporting potentially misleading accuracy values), improving the writing in the paper, rectifying and adding some related works, and providing additional experimental plots for more detailed analysis, and various added details in the appendix.

During the discussion phase, one reviewer has indicated that parts of their concerns were indeed addressed, in particular due to the addition of membership inference experimentation, yet another has noted in the internal AC-reviewer discussion that improvements were made but the big points remain unaddressed. As such, the large spread of ratings persisted in the final scores. This lead to the AC reading the full paper and following the discussion carefully.

Ultimately, the AC unfortunately agrees with the reviewers who point out that the paper has indeed improved already and is moving into the right direction, yet requires more revision before being publishable. A more precise set of reasons is provided in the justification box below.

**Justification For Why Not Higher Score:**

The decision to reject this paper was not taken lightly. In fact, the AC acknowledges that an incredible attempt at including significant improvements has been made within the rather short discussion period.

However, after carefully reading the paper, the AC believes that a more thorough revision of the paper would be to substantial benefit to the paper, resulting in a much improved final version that what is presented right now. The precise points, raised in large parts also by the reviewers, are as follows:

* The introduction is very long and seems to be in parts tangential to the class unlearning set-up discussed earlier. In the rebuttal, the authors conjecture on potentially realistic settings in e.g. unlearning of individuals (or groups of individuals) in face recognition, yet the demonstrated CIFAR and ImageNet experiments, despite scale, do not really fit the motivation well. The AC recommends clarifying the motivation, shortening the introduction (see points below), or adding a mentioned related example, e.g. of removing groups of individuals, to the experiments.
*  Whereas the method is intuitive and simple, some of the motivation behind the parameters is lacking. Reviewers have noted that they are concerned about the lack of either theoretical guarantees and empirical intuition behind the parameters. The rebuttal makes an attempt at clarifying that scaling is regularly used in continual learning and the paper mentions that the two alphas can be found through grid search. There is no mention or analysis on the effect of the chosen data subsets for retain and forget, which should have a rather dramatic influence on the proposed technique (similar to how core set size is the primary influencing factor for forgetting in continual learning). The AC recommends including a more thorough motivation and rigorous empirical analysis behind these choices.
* Despite the experimental evidence being the first to e.g. consider vision transformer architectures, the current empirical analysis is still limited. Reviewers mentioned that accuracy values alone do not provide sufficient evidence and the authors have taken the first steps to include membership inference attacks and decompose reported values into a broader set of figures (e.g. including one example for parameter differences and a single qualitative visualization of GradCAM). These are great steps to strengthen the paper significantly, yet in the current version are too short and came at the expense of clarity in the paper. In particular, the entire experimental section and discussion is very cramped and very hard to follow as a reader. The AC imagines this to be an effect of the short period to make the changes, and thus believes that spending significantly more time for a major revision with more such experiments and discussion will be to the ultimate benefit of the paper. This may require some restructuring of the paper, for instance shortening the 2.5 page introduction significantly, such that all relevant details are not deferred to the appendix. The latter should be supplementary in nature, and not be required for basic understanding of the paper, which unfortunately is not the case right now.

In the end, as mentioned above, the paper is making great initial steps based on a novel and prospectively important idea, which is why it is likely to have a much better future impact if the paper is adequately revised first. The AC hopes the authors take all constructive feedback into account carefully in a future resubmission.

**Justification For Why Not Lower Score:**

N/A

---

### Decision · Program_Chairs · 2024-01-16

Reject